# Phenoxyaromatic Acid Analogues as Novel Radiotherapy Sensitizers: Design, Synthesis and Biological Evaluation

**DOI:** 10.3390/molecules27082428

**Published:** 2022-04-09

**Authors:** Hongquan Zhang, Chunxi Wen, Bingting Li, Xinlin Yan, Yangrong Xu, Jialin Guo, Shi Hou, Jiajia Chang, Song Li, Junhai Xiao

**Affiliations:** 1National Engineering Research Center for the Emergency Drug, Beijing Institute of Pharmacology and Toxicology, Beijing 100850, China; zhanghqmlj@126.com (H.Z.); wenchunxi1996@126.com (C.W.); yanxinlin11@163.com (X.Y.); yrxuytu@163.com (Y.X.); shandongguojialin@163.com (J.G.); houshi28@126.com (S.H.); jiajiachang@126.com (J.C.); lis@nic.bmi.ac.cn (S.L.); 2State Key Laboratory of Toxicology and Medical Countermeasures, Beijing Institute of Pharmacology and Toxicology, Beijing 100850, China; 3School of Pharmaceutical Sciences, Jilin University, Changchun 130021, China; 4Institute of Health Service and Transfusion Medicine, Beijing 100850, China; libingtingvvvip@163.com

**Keywords:** phenoxyacetic acid, hemoglobin, tumor cell, radiotherapy sensitizers, molecular docking

## Abstract

Radiotherapy is a vital approach for brain tumor treatment. The standard treatment for glioblastoma (GB) is maximal surgical resection combined with radiotherapy and chemotherapy. However, the non-sensitivity of tumor cells in the hypoxic area of solid tumors to radiotherapy may cause radioresistance. Therefore, radiotherapy sensitizers that increase the oxygen concentration within the tumor are promising for increasing the effectiveness of radiation. Inspired by hemoglobin allosteric oxygen release regulators, a series of novel phenoxyacetic acid analogues were designed and synthesized. A numerical method was applied to determine the activity and safety of newly synthesized compounds. In vitro studies on the evaluation of red blood cells revealed that compounds **19c** (∆P_50_ = 45.50 mmHg) and **19t** (∆P_50_ = 44.38 mmHg) improve the oxygen-releasing property effectively compared to positive control efaproxiral (∆P_50_ = 36.40 mmHg). Preliminary safety evaluation revealed that **19c** exhibited no cytotoxicity towards HEK293 and U87MG cells, while **19t** was cytotoxic toward both cells with no selectivity. An in vivo activity assay confirmed that **19c** exhibited a radiosensitization effect on orthotopically transplanted GB in mouse brains. Moreover, a pharmacokinetic study in rats showed that **19c** was orally available.

## 1. Introduction

Hemoglobin, an essential oxygen carrier, is composed of four iron-containing subunits. Stereo conformational isomerism of hemoglobin affects its oxygen affinity, while allosteric modulators change the oxygen affinity of red blood cells by altering the three-dimensional conformation of hemoglobin [1]. Hemoglobin appears to be in a tense state (T state) in the deoxygenated form, which is conducive to the release of oxygen molecules. Alternatively, the oxygenated hemoglobin is in a relaxed state (R state), which tends to bind oxygen molecules [2,3]. The development of a series of hemoglobin allosteric effectors will be beneficial in the treatment of diseases linked to oxygen supply.

GB is one of the most malignant tumors, with a median survival time of 12–15 months [4,5,6,7]. The standard treatment for GB is maximal surgical resection combined with radiotherapy and chemotherapy [8,9,10]. However, the blood–brain barrier (BBB) prevents small molecules from entering tumor tissues to exert their pharmacological effects [11,12,13]. As a result, the necessity of BBB passage complicates the design of chemotherapeutic drugs [14,15,16]. However, allosteric modulators of hemoglobin are unaffected by the obstacles of the BBB. They have the ability to increase the oxygen supply to the hypoxic area of the tumor, producing a sensitization to radiotherapy [17].

Because of the angiogenesis abnormalities during the growth of tumor tissues and vigorous growth of tumor cells, the oxygen supply to tumor tissues is insufficient [18,19]. As a result, many types of tumors have hypoxic areas [20,21,22]. Tumor cells in the hypoxic area of the tumor and the tumor cells in the normoxic area respond differently to treatment due to varied sensitivity [23,24]. The existence of the tumor hypoxic zone is one of the main causes for resistance to radiotherapy.

Additionally, because of the potential of hemoglobin allosteric modulators in tissue oxygen supply, scientists have developed a series of hemoglobin allosteric modulators [25,26,27]. Among them, organophosphate compounds represented by inositol tripyrophosphate (ITPP) have shown potent allosteric effects on the T state of hemoglobin. However, it is difficult for ITPP to penetrate the cell membrane of red blood cells, limiting the development of such compounds [28]. Phenoxyacetic acid compounds were initially identified as lipid-lowering drugs. Later, Perutz and Abraham found that clofibrate and bezafibrate can stabilize the T state hemoglobin, leading to the development of a series of phenoxyacetic acid analogues, of which the most promising compound is efaproxiral [29,30,31,32]. A phase III clinical trial has shown that efaproxiral can enhance the effect of radiotherapy in patients with advanced lung cancer [33]. However, due to the relatively weak allosteric effect of efaproxiral, a relatively large dose is needed in clinical practice to achieve a therapeutic effect. Additionally, its long-term usage may produce dose-related side effects [34]. Therefore, based on the properties of hemoglobin allosteric modulators, a more effective phenoxyacetic acid derivative is needed for solid tumor radiotherapy, besides efaproxiral.

In this study, a series of new phenoxyacetic acid analogues were designed and synthesized based on structure–activity relationships (SARs) using molecular docking tools. In vitro activity evaluation proved that halogenation, aromatic ring substitution, or thioether substitution of the hydrophobic region is crucial for improving compound activity. Moreover, a suitable linker length in the hinge region is essential. Herein, we report the discovery of a few potential phenoxyacetic acid analogues, where their allosteric modulation effect was studied in vitro. Moreover, our findings were verified by molecular docking. The radiosensitization effect of **19c** was confirmed in vivo in glioma bearing mice.

## 2. Results and Discussion

### 2.1. Design and Syntheses of Phenoxyaromatic Acid Analogues

As is shown in Figure 1a, the reported hemoglobin allosteric effectors shared some common features. Most of these structures possess a biaryl scaffold with an amide-bound linker (Figure 1b). The terminal region is a methylpropionic acid or methylpropionic acid ester tail.

Based on the binding mode of efaproxiral (Figure 2), the SARs of the reported phenoxyaromatic acid derivatives were analyzed. Interestingly, the 3,5-dimethyl aniline moiety of efaproxiral binds in the hydrophobic pocket formed by residues Val96, Lys99, Leu100, His103, Phe36, and Asn108 [35]. We can thus interpret that the substitution of 3,5-dimethyl aniline with proper hydrophobic groups may maintain or even improve its activity. The carbonyl oxygen of the effectors formed a direct hydrogen bond with the Lys99 residue, which is essential to stabilize the T state conformation of deoxyhemoglobin. As a result, it was retained in a position similar to that in many of our newly designed compounds. As observed in Figure 2b, the methylpropionic acid tail of efaproxiral formed a few water-mediated hydrogen bonds with Lys99 and Arg541 in the central water cavity. Therefore, we designed various analogues with linker and tail region replacement to obtain stronger and more direct contact in the cavity. Inspired by all of these SAR results, we designed three series of compounds including the substitution of the benzene ring in the head region (region I) along with exploration of the linker structure (region II) and terminal region replacement (region III), as shown in Figure 1c.

The syntheses of key intermediates **12a–12y** are outlined in Figure 1. The amide condensation between substituted aniline (**10a-r**) and p-hydroxybenzoic acid (**11a-h**) was carried out with the addition of a condensing agent EDCl, HOBT to obtain **12b**, **12f–12w** or with the addition of catalyst boric acid to produce **12a**, **12c–12e** and **12x–12y**.

The syntheses of effectors **15a**–**d**, **16a**–**c**, **18a**–**g**, **19a**–**x** and **20** are outlined in Figure 2. The O-alkylation of **12a** with **14a**–**d** yielded **15a–d** [36]. Effectors **16a**–**c** were obtained from **15a**–**c** by ester hydrolysis under alkaline conditions. As described in Figure 2, **17a**–**g** were reduced from **12a**, **12c**, **12e**, **12j**, **12m-1**, **12n-1** and **12p,** respectively, using LiAlH_4_, and were transformed to desired compounds **18a-g** by Bargellini reaction [37]. The Bargellini reaction was used to obtain desired compounds **19a**–**x** from intermediates **12a**–**x**. The O-alkylation of **12y** with 2,3-dibromopropionic acid yielded **20**.

The syntheses of **24a**–**h** are shown in Figure 3. Intermediates **23a**–**h** were obtained by amide condensation between substituted amino **22a**–**h** and 4-hydroxyphenylacetic acid. Target molecules **24a**–**h** were provided by Bargellini reaction of **23a**–**h**.

### 2.2. In Vitro Red Blood Cell Evaluation and SAR Analysis

In the first stage of our research, three series of analogues were designed and synthesized to explore refinement of the hydrophobic parts of **19f**, **19m**, **24a** and **24b**; hinge areas of **18g**, **19b** and **19y**; and tail regions of **15c**, **15d** and **16c,** respectively. We used the Softron Analyzer to evaluate the effect of these compounds on the half-blood oxygen saturation(P_50_) value of red blood cells. The most promising phenoxyaromatic acid compound, efaproxiral, was used as a positive control. As is shown in Table 1, Table 2 and Table 3, compound **18g** (∆P_50_ = 35.71 mmHg), **19m** (∆P_50_ = 33.05 mmHg) and **19y** (∆P_50_ = 38.78 mmHg) exhibited a similar effect compared to efaproxiral (∆P_50_ = 36.40 mmHg). The results reminded us that the modification of the hydrophobic part or linker extension is beneficial to improve an allosteric effect.

Next, a total of 26 compounds were designed and synthesized. Further evaluation showed that **19i** (∆P_50_ = 41.37 mmHg), **19m-1** (∆P_50_ = 36.43 mmHg), **19n-1** (∆P_50_ = 38.73 mmHg), **24d** (∆P_50_ = 38.31 mmHg) and **19c** (∆P_50_ = 45.50 mmHg) were as good as or even better than efaproxiral; of these, **19c (**∆P_50_ = 45.50 mmHg) had the highest ∆P_50_ (Table 2). In vitro analysis revealed that the most successful exploration of the linker part was the chain structure composed of three carbon atoms (Table 2). Moreover, the halogenation or introduction of a group that could increase the conjugated system of the benzene ring of the hydrophobic part increased the activity of the effectors (Table 1, Table 2 and Table 3). Compared with para substitution products **19o** (∆P_50_ = 33.37 mmHg), **19m** (∆P_50_ = 33.05 mmHg) and**19n** (∆P_50_ = 19.15 mmHg), meta substitution products **19p** (∆P_50_ = 35.91 mmHg), **19m-1** (∆P_50_ = 36.43 mmHg) and **19n-1** (∆P_50_ = 38.73 mmHg) were more effective (Table 1).



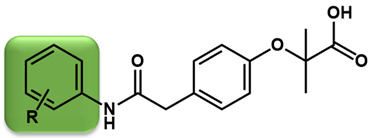




**Series I.**


**Table 1 molecules-27-02428-t001:** Results of in vitro red blood cell studies.

Compound	Region I	P_50_ (mmHg)	∆P_50_ (mmHg)	P_50_/P_50_C	K
**19f**	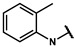	53.78 ± 1.94	7.72 ± 1.87	1.17 ± 0.04	2.26 ± 0.11
**19g**	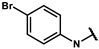	71.97 ± 0.49	31.92 ± 1.14	1.80 ± 0.06	2.20 ± 0.20
**19h**	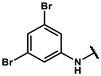	68.71 ± 1.18	31.90 ± 1.18	1.87 ± 0.03	2.41 ± 0.01
**19i**	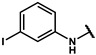	86.61 ± 6.04	41.37 ± 6.04	1.91 ± 0.13	2.47 ± 0.13
**19j**	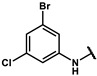	82.35 ± 0.80	37.11 ± 0.80	1.82 ± 0.02	1.82 ± 0.33
**19k**	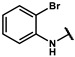	49.17 ± 10.7	12.36 ± 10.79	1.34 ± 0.29	2.30 ± 0.08
**19l**	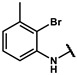	70.41 ± 1.68	25.17 ± 1.68	1.56 ± 0.04	2.60 ± 0.04
**19m**	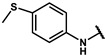	74.24 ± 0.69	33.05 ± 0.69	1.80 ± 0.02	2.17 ± 0.09
**19m-1**	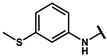	81.67 ± 0.95	36.43 ± 0.95	1.81 ± 0.02	2.55 ± 0.06
**19n**	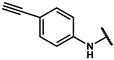	60.34 ± 0.70	19.15 ± 0.70	1.46 ± 0.02	2.49 ± 0.06
**19n-1**	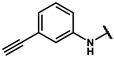	80.97 ± 1.77	38.73 ± 2.47	1.79 ± 0.04	2.38 ± 0.19
**190**	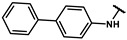	74.56 ± 1.75	33.37 ± 1.75	1.81 ± 0.04	2.42 ± 0.21
**19p**	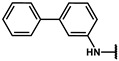	77.10 ± 0.66	35.91 ± 0.66	1.87 ± 0.02	1.77 ± 0.10
**19q**	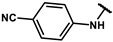	59.60 ± 13.58	14.36 ± 13.58	1.32 ± 0.30	2.42 ± 0.11
**19r**	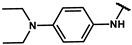	65.81 ± 6.64	23.97 ± 1.82	1.57 ± 0.02	2.54 ± 0.03
**19s**	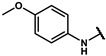	71.87 ± 0.64	22.29 ± 0.64	1.45 ± 0.01	2.53 ± 0.16
**24a**	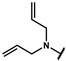	55.80 ± 18.99	11.80 ± 11.11	1.25 ± 0.21	2.70 ± 0.27
**24b**	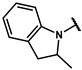	66.26 ± 5.30	21.02 ± 5.30	1.46 ± 0.12	2.63 ± 0.11
**24c**	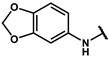	69.62 ± 12.16	25.61 ± 4.27	1.58 ± 0.01	2.63 ± 0.09
**24d**	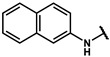	79.50 ± 0.47	38.31 ± 0.47	1.93 ± 0.01	2.41 ± 0.06
**24e**	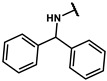	59.61 ± 20.96	15.61 ± 13.07	1.33 ± 0.24	2.67 ± 0.08
**24f**	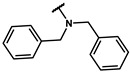	54.51 ± 14.91	10.51 ± 7.02	1.23 ± 0.12	2.61 ± 0.12
**24g**	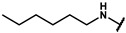	46.17 ± 7.11	4.34 ± 2.30	1.10 ± 0.04	2.38 ± 0.12
**24h**	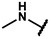	58.11 ± 1.93	16.27 ± 2.88	1.40 ± 0.11	2.10 ± 0.21

Red blood cell analyses were carried out at a final effector concentration of 2 mM. All stock solutions were prepared in DMSO at a concentration of 200 mM. P_50_ represents the partial pressure of oxygen when 50% hemoglobin is saturated in the presence of effectors. P_50_C is the control value of P_50_ in the presence of 1% DMSO. ∆P_50_ = (P_50_ − P_50_C) in mmHg. K is the Hill coefficient of the oxygen dissociation curve when the red blood cells are 50% saturated. Data are displayed as mean ± SD.



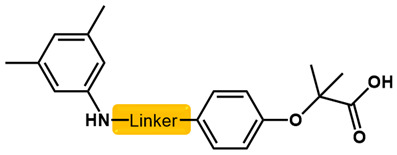




**Series II**


**Table 2 molecules-27-02428-t002:** Results of in vitro red blood cell studies.

Compound	Region II	P_50_ (mmHg)	∆P_50_ (mmHg)	P_50_/P_50_C	K
**Efaproxiral** **(** **19a** **)**	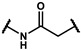	82.50 ± 1.01	36.40 ± 1.01	1.79 ± 0.02	2.10 ± 0.04
**19b**	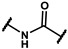	43.01 ± 2.18	1.82 ± 2.18	1.04 ± 0.05	2.04 ± 0.42
**19c**	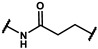	89.16 ± 5.20	45.50 ± 1.73	2.04 ± 0.04	2.43 ± 0.12
**19d**	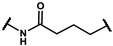	67.93 ± 8.80	25.43 ± 3.71	1.60 ± 0.02	2.06 ± 0.19
**19e**	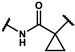	46.24 ± 7.09	4.41 ± 2.27	1.10 ± 0.04	2.40 ± 0.12
**19y**	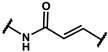	84.02 ± 2.64	38.78 ± 2.64	1.86 ± 0.06	2.54 ± 0.23
**18a**	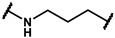	77.19 ± 5.93	22.54 ± 2.74	1.41 ± 0.03	2.36 ± 0.21
**18g**	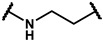	77.54 ± 6.96	35.71 ± 2.14	1.86 ± 0.05	1.74 ± 0.35

Red blood cell analyses were carried out at a final effector concentration of 2 mM. All stock solutions were prepared in DMSO at a concentration of 200 mM. P_50_ represents the partial pressure of oxygen when 50% hemoglobin is saturated in the presence of effectors. P_50_C is the control value of P_50_ in the presence of 1% DMSO. ∆P_50_ = (P_50_ − P_50_C) in mmHg. K is the Hill coefficient of the oxygen dissociation curve when the red blood cells are 50% saturated. Data are displayed as mean ± SD.



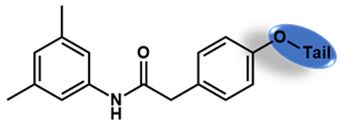




**Series III**


**Table 3 molecules-27-02428-t003:** Results of in vitro red blood cell studies.

Compound	Region III	P_50_(mmHg)	∆P_50_ (mmHg)	P_50_/P_50_C	K
**15a**	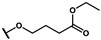	43.55 ± 11.21	−1.70 ± 11.21	0.96 ± 0.25	2.31 ± 0.11
**15b**	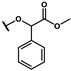	57.82 ± 8.69	12.58 ± 8.69	1.28 ± 0.19	1.97 ± 0.74
**15c**	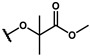	43.28 ± 11.71	−1.96 ± 11.71	0.96 ± 0.26	2.29 ± 0.14
**16a**	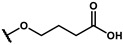	53.49 ± 1.85	8.25 ± 1.85	1.18 ± 0.04	2.40 ± 0.04
**20**	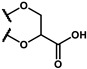	47.11 ± 8.08	5.27 ± 3.27	1.12 ± 0.06	2.39 ± 0.16

Red blood cell analyses were carried out at a final effector concentration of 2 mM. All stock solutions were prepared in DMSO at a concentration of 200 mM. P_50_ represents the partial pressure of oxygen when 50% hemoglobin is saturated in the presence of effectors. P_50_C is the control value of P_50_ in the presence of 1% DMSO. ∆P_50_ = (P_50_ − P_50_C) in mmHg. K is the Hill coefficient of the oxygen dissociation curve when the red blood cells are 50% saturated. Data are displayed as mean ± SD.

Finally, we combined the selected hydrophobic structure and linker structure obtained in the first two stages for better effectors. In vitro evaluation showed that most of the compounds designed by this strategy produced negative results. However, we obtained a potential compound, **19t** (∆P_50_ = 44.38 mmHg), with a robust allosteric effect as well (Table 4).



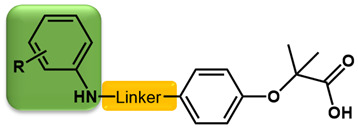




**Series IV**


**Table 4 molecules-27-02428-t004:** Results of in vitro red blood cell studies.

Compound	Region I & II	P_50_ (mmHg)	∆P_50_ (mmHg)	P_50_/P_50_C	K
**18b**	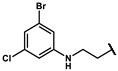	42.61 ± 2.15	3.66 ± 0.88	1.10 ± 0.03	2.19 ± 0.01
**18c**	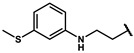	48.61 ± 4.43	10.7 ± 2.95	1.28 ± 0.07	2.01 ± 0.22
**18d**	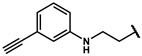	52.25 ± 2.32	7.01 ± 2.32	1.15 ± 0.05	2.25 ± 0.04
**18e**	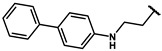	51.84 ± 16.79	6.60 ± 16.79	1.15 ± 0.37	2.28 ± 0.28
**18f**	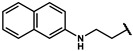	79.24 ± 1.61	33.13 ± 1.61	1.72 ± 0.03	1.89 ± 0.13
**19t**	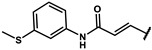	89.62 ± 2.02	44.38 ± 2.02	2.13 ± 0.07	2.43 ± 0.16
**19u**	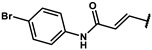	63.15 ± 3.48	25.24 ± 3.48	1.67 ± 0.09	1.96 ± 0.17
**19v**	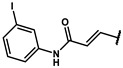	77.49 ± 2.11	32.25 ± 2.11	1.71 ± 0.05	2.23 ± 0.28
**19w**	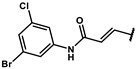	56.76 ± 2.93	18.85 ± 2.93	1.50 ± 0.08	1.79 ± 0.01

Red blood cell analyses were carried out at a final effector concentration of 2 mM. All stock solutions were prepared in DMSO at a concentration of 200 mM. P_50_ represents the partial pressure of oxygen when 50% hemoglobin is saturated in the presence of effectors. P_50_C is the control value of P_50_ in the presence of 1% DMSO. ∆P_50_ = (P_50_ − P_50_C) in mmHg. K is the Hill coefficient of the oxygen dissociation curve when the red blood cells are 50% saturated. Data are displayed as mean ± SD.

### 2.3. Analysis of Cytotoxicity for Preliminary Safety Evaluation

For further in vivo evaluation of newly synthesized compounds, **19c** and **19t** were selected as candidates. Hemoglobin allosteric effectors require a relatively high blood concentration to be effective. Cytotoxicity evaluation was conducted out of consideration for medication safety. HEK293 and U87MG cells were used to evaluate the cytotoxicity. As revealed in Figure 3, **19t** was cytotoxic to both HEK293 cells (TC_50_ = 165 μM) and U87MG cells (IC_50_ = 183 μM) without selectivity (selection index = 1.1). Compound 19t showed cytotoxicity to HEK293 and U87MG cells at a concentration of 10 μm. Meanwhile, there was no cytotoxic effect observed for **19c** on the HEK293 and U87MG cells at a concentration of 5 mM.

### 2.4. The Influence of Compound ***19c*** on the Bohr Effect of Red Blood Cells and the Analysis of Its Dose–Response Relationship

The existence of the Bohr effect preserves the regulation of hemoglobin oxygen affinity by endogenous allosteric agents 2,3 DPG and Cl^−^. Therefore, it is essential to illustrate whether or not it still exists in the presence of an exogenous effector. As a result, the influence of **19c** on the Bohr effect of mouse red blood cells was analyzed. The results showed that a change in acidity would lead to an apparent oxygen equilibrium curve (OEC) shift (Figure 4a). This phenomenon indicates that **19c** will not offset the Bohr effect of hemoglobin.

Research on the in vitro dose–response relationship of **19c** is essential for evaluating the potential of this compound. It will provide crucial support for in vivo antitumor evaluation. As shown in Figure 4b, compound **19c** dose-dependently altered the oxygen affinity of hemoglobin. This suggests that in the in vivo radiosensitization experiments, the oxygen partial pressure in the hypoxic region of the tumor can be improved by increasing the dose of **19c**.

### 2.5. Docking Results

To illustrate the differences between **19c** and efaproxiral at a molecular level, the docking results were carefully analyzed. Compared to efaproxiral (Figure 2), linker extension changed the binding mode of methylpropionic acid tail in the central water cavity. Compound **19c** formed more water-mediated hydrogen bounds, and it preserved all binding positions in the hydrophobic region (Figure 5b). Compared to efaproxiral, **19c** fitted deeper into the central water cavity and formed water-mediated hydrogen bonds with Lys127, Thr534, Thr537, Ser538 and Thr540 as well as Arg541. These results explained the increased allosteric activity of **19c** (Figure 5a).

### 2.6. In Vivo Radiosensitization Effect on Brain Tumor

In order to evaluate in vivo antitumor effects, luciferase-labeled U87MG was implanted into Balb/c nude mice brains. A small animal bioluminescence imaging system was applied to trace the tumor formation and dynamic changes in GB size. Cobalt 60 was used for fractionated-dose whole-brain irradiation therapy (IR). The tumor was irradiated with a total dose of 12 Gy in four fractions of 3 Gy at a dose rate of 0.3 Gy/min. Efaproxiral or **19c** was administrated at a dosage of 150 mg/kg by intraperitoneal injection (ip) every four days over a period of sixteen days. IR was conducted 15 min after ip.

As shown in (Figure 6a,b), compared to vehicle group, Efa + IR, IR and **19c** + IR all significantly inhibited tumor proliferation with *p* values of 0.0270, 0.0255 and 0.0211, respectively. Compared to IR alone, **19c** combined with IR significantly inhibited tumor progression (*p* = 0.0449); however, efaproxiral combined with IR was statistically ineffective compared with IR therapy alone (*p* = 0.3462). Consistent with Figure 6, the results for the survival period of tumor-bearing mice showed that IR alone or combined with **19c** or efaproxiral could improve the overall survival time of tumor-bearing mice compared to vehicle control (Figure 7a,b). The survival time obtained with **19c** combined with IR was significantly longer than that obtained with IR alone (*p* = 0.0286). Since we did not directly detect the changes in oxygenation in the tumor area after compound **19c** administration during the in vivo radiosensitization experiments, in order to better illustrate the radiosensitization effect of **19c**, we prepared a hypothetical illustration. The hypothesized radiosensitization mechanism of **19c** is shown in Figure 6c. As shown in the schematic diagram, **19c** administration would increase tumor oxygenation in GB. Reoxygenation of the hypoxic area sensitized IR treatment.

### 2.7. Preliminary Pharmacokinetic Assessment

Based on the potential radiosensitization effects in vivo, we performed an in vivo pharmacokinetic (PK) study to preliminarily evaluate the druggability of **19c** at a dosage of 100 mg/kg by intraperitoneal injection (ip) or oral administration (po). The corresponding PK parameters are summarized in Table 5. After ip or po administration of **19c**, the values of AUCinf were 612,428 ng/mL or 410,402 ng/mL, respectively, which indicated moderate systemic exposure. In addition, **19c** showed a half-life (T1/2) of 2.55 h and 2.78 h, respectively. These results indicated that **19c** showed moderate PK properties via ip or po, which provided a foundation for further development. As is shown in Appendix A, the plasma concentration of **19c** after oral administration was more stable than that after intraperitoneal injection, and the pharmacokinetic parameters were similar to those of intraperitoneal injection, which suggests that oral administration may be preferred.

## 3. Conclusions

GB is one of the most lethal tumors. However, due to the hindrance of the BBB, drug development has lagged [38,39,40]. It is particularly important to overcome radioresistance influenced by the presence of hypoxia. The existing research shows that phenoxyacetic acid derivatives represented by efaproxiral will increase the tumor oxygen supply. Thus, a series of novel phenoxyacetic acid derivatives were designed, synthesized, and evaluated. The SAR results demonstrated that a linker of three carbon atoms in the hinge area is the optimum structure for phenoxyacetic acid derivatization. Moreover, thioether and aromatic ring substitution of the hydrophobic region was shown to improve the activity of the newly synthesized compounds.

Additionally, the activity of a meta-substituted benzene ring in the hydrophobic region is better than that achieved through para-substitution. In the in vitro activity evaluation stage, we confirmed that both 19c and 19t induced in vitro hemoglobin allosteric activities that were significantly higher than that of the positive control. However, a preliminary safety evaluation revealed that 19t was cytotoxic to both normal and glioma cells at low micromolar levels without selectivity. Furthermore, in vitro activity evaluation showed that **19c** exerts an allosteric effect on hemoglobin at a micromolar level. Moreover, **19c** preserved the Bohr effect of hemoglobin, which means it will not interfere with the endogenous hemoglobin effectors. The in vivo antitumor results indicated that **19c** is an effective GB radiosensitizer.

In conclusion, our preliminary work confirmed that **19c** is an effective radiosensitizer. To obtain more potent effectors, further optimization of compound **19c** based on SAR analysis is under investigation.

## 4. Materials and Methods

### 4.1. General Chemistry

All solvents and reagents were obtained from commercial suppliers and used directly. Silica gel TLC plates (GF254) were applied to monitor the reactions. Silica gel (200−300 mesh) was used for chromatography. The ^1^H and ^13^C NMR spectra were recorded on a JEOLECA400 spectrometer, with TMS as an internal standard at ambient temperature. All chemical shifts are reported in parts per million (ppm). All coupling constants are reported in hertz. The ESI-MS was recorded on an Agilent TOF G6230A mass spectrometer.

### 4.2. Procedure for the Syntheses of Intermediates ***12a–12z***

#### 4.2.1. N-(3,5-dimethylphenyl)-2-(4-hydroxyphenyl)acetamide (**12a**)

To a stirring solution of 3,5-dimethylaniline **10a** (2.7 g, 15.6 mmol) in toluene (25 mL) were add 2-(4-hydroxyphenyl)acetic acid **11a** (2.0 g, 13 mmol) and boric acid (0.08 g, 0.13 mmol) at room temperature. The reaction mixture was refluxed at 140℃ with a water separator for 12 h. Upon completion, the mixture was cooled to room temperature and filtered, and the filter cake was washed with toluene (3 × 20 mL) and water (3 × 20 mL), respectively. Then, the filter cake was dried in a vacuum drying oven to give **12a** as a white solid (3.0g, 90.5%). ^1^H NMR (400 MHz, DMSO-*d6*) δ 9.88 (s, 1H), 9.25–9.23 (m, 1H), 7.18 (d, *J* = 0.6 Hz, 2H), 7.10–7.05 (m, 2H), 6.69–6.64 (m, 2H), 6.63 (dd, *J* = 4.2, 3.6 Hz, 1H), 3.42 (d, *J* = 5.6 Hz, 2H), 2.18 (d, *J* = 0.5 Hz, 6H). ESI-MS: *m*/*z* = 256.11 [M + H]^+^.

#### 4.2.2. N-(3,5-dimethylphenyl)-4-hydroxybenzamide (**12b**)

To a stirring solution of 4-hydroxybenzoic acid **11b** (0.73 g, 6 mmol), EDCl (1.44 g, 7.5 mmol) and HOBT (0.81 g, 6 mmol) in dry DCM (20 mL) at room temperature was added 3,5-dimethylaniline **10a** (0.73 g, 6 mmol) for 8 h. Upon completion, the mixture was washed with water (3 × 20 mL) and dried over anhydrous sodium sulfate before concentration in vacuo. The crude product was purified with silica gel column chromatography using petroleum ether (PE)/ethyl acetate (EA) (5:1 to 3:1) as eluent to afford **12b** as a white solid (0.53 g, 44%). ^1^H NMR (600 MHz, DMSO-*d6*) δ 10.05 (s, 1H), 9.80 (s, 1H), 7.85–7.82 (m, 2H), 7.39 (s, 2H), 6.87–6.83 (m, 2H), 6.71 (s, 1H), 2.25 (s, 6H). ESI-MS: *m*/*z* = 2542.11 [M + H]^+^.

#### 4.2.3. N-(3,5-dimethylphenyl)-3-(4-hydroxyphenyl)propenamide (**12c**)

The title compound was obtained similarly to **12a**. 2-(4-hydroxyphenyl)acetic acid **11a** was replaced with 3-(4-hydroxyphenyl)propanoic acid **11c.** A pale white solid was obtained; yield: 87.8%. ^1^H NMR (600 MHz, DMSO-*d6*) δ 9.69 (s, 1H), 9.14 (s, 1H), 7.20 (s, 2H), 7.02 (d, *J* = 8.3 Hz, 2H), 6.70–6.61 (m, 3H), 2.77 (t, *J* = 7.7 Hz, 2H), 2.56–2.51 (m, 2H), 2.21 (s, 6H). ESI-MS: *m*/*z* = 270.15 [M + H]^+^.

#### 4.2.4. N-(3,5-dimethylphenyl)-4-(4-hydroxyphenyl)butanamide (**12d**)

The title compound was obtained similarly to **12a**. 2-(4-hydroxyphenyl)acetic acid **11a** was replaced with 4-(4-hydroxyphenyl)butanoic acid **11d.** A pale white solid was obtained; yield: 89.0%. ^1^H NMR (400 MHz, DMSO-*d6*) δ 9.66 (s, 1H), 9.10 (s, 1H), 7.17 (s, 2H), 6.93 (dd, *J* = 8.8, 6.2 Hz, 2H), 6.63 (dt, *J* = 4.0, 2.4 Hz, 3H), 2.45–2.41 (m, 2H), 2.22 (t, *J* = 7.5 Hz, 2H), 2.17 (s, 6H), 1.81–1.72 (m, 2H). ESI-MS: *m*/*z* = 284.15 [M + H]^+^.

#### 4.2.5. N-(3,5-dimethylphenyl)-1-(4-hydroxyphenyl)cyclopropane-1-carboxamide (**12e**)

The title compound was obtained similarly to **12b.** 2-(4-hydroxyphenyl)acetic acid **11a** was replaced with 1-(4-hydroxyphenyl)cyclopropane-1-carboxylic acid **11e.** A pale yellow solid was obtained; yield: 68.6%. ^1^H NMR (400 MHz, DMSO-*d6*) δ 9.43 (s, 1H), 8.29 (s, 1H), 7.20–7.14 (m, 2H), 7.06 (d, *J* = 0.7 Hz, 2H), 6.75–6.69 (m, 2H), 6.63–6.60 (m, 1H), 2.14 (d, *J* = 0.5 Hz, 6H), 1.32 (dd, *J* = 6.6, 3.8 Hz, 2H), 0.95 (dd, *J* = 6.8, 4.0 Hz, 2H). ESI-MS: *m*/*z* = 282.15 [M + H]^+^.

#### 4.2.6. 2-(4-hydroxyphenyl)-N-(o-tolyl)acetamide (**12f**)

The title compound was obtained similarly to **12a**. 3,5-dimethylaniline **10a** was replaced with o-toluidine **10b.** A pale white solid was obtained; yield: 87.0%. ^1^H NMR (400 MHz, DMSO-*d6*) δ 9.34 (s, 1H), 9.24 (s, 1H), 7.35–7.31 (m, 1H), 7.13 (dd, *J* = 15.6, 8.0 Hz, 4H), 7.03 (td, *J* = 7.3, 1.2 Hz, 1H), 6.71–6.66 (m, 2H), 3.49 (s, 2H), 2.11 (s, 3H). ESI-MS: *m*/*z* = 242.10 [M + H]^+^.

#### 4.2.7. N-(4-bromophenyl)-2-(4-hydroxyphenyl)acetamide (**12g**)

The title compound was obtained similarly to **12a**. 3,5-dimethylaniline **10a** was replaced with 4-bromoaniline **10c.** A brown solid was obtained; yield: 81.8%.^1^H NMR (400 MHz, DMSO-*d6*) δ 10.19 (s, 1H), 9.25 (s, 1H), 7.57–7.51 (m, 2H), 7.47–7.41 (m, 2H), 7.10–7.03 (m, 2H), 6.70–6.64 (m, 2H), 3.46 (s, 2H). ESI-MS: *m*/*z* = 305.99 [M + H]^+^.

#### 4.2.8. N-(3,5-dibromophenyl)-2-(4-hydroxyphenyl)acetamide (**12h**)

The title compound was obtained similarly to **12a**. 3,5-dimethylaniline **10a** was replaced with 3,5-dibromoaniline **10d.** A brown solid was obtained; yield: 44.0%. ^1^H NMR (400 MHz, DMSO-*d6*) δ 10.35 (s, 1H), 9.27 (s, 1H), 7.81 (d, *J* = 1.7 Hz, 2H), 7.45 (t, *J* = 1.7 Hz, 1H), 7.09–7.04 (m, 2H), 6.69–6.64 (m, 2H), 3.47 (s, 2H). ESI-MS: *m*/*z* = 385.90 [M + H]^+^.

#### 4.2.9. 2-(4-hydroxyphenyl)-N-(4-iodophenyl)acetamide (**12i**)

The title compound was obtained similarly to **12b.** 3,5-dimethylaniline **10a** was replaced with 3-iodoaniline **10e.** A pale purple solid was obtained; yield: 45.3%. ^1^H NMR (400 MHz, DMSO-*d6*) δ 10.18 (s, 1H), 9.30 (s, 1H), 8.06 (t, *J* = 1.8 Hz, 1H), 7.51–7.45 (m, 1H), 7.37–7.31 (m, 1H), 7.08–7.01 (m, 3H), 6.69–6.61 (m, 2H), 3.44 (s, 2H). ESI-MS: *m*/*z* = 354.00 [M + H]^+^.

#### 4.2.10. N-(3-bromo-5-chlorophenyl)-2-(4-hydroxyphenyl)acetamide (**12j**)

The title compound was obtained similarly to **12a**. 3,5-dimethylaniline **10a** was replaced with 3-bromo-5-chloroaniline **10f.** A brown solid was obtained; yield: 92.5%. ^1^H NMR (400 MHz, DMSO-*d6*) δ 10.37 (s, 1H), 9.27 (s, 1H), 7.76 (t, *J* = 1.8 Hz, 1H), 7.66 (t, *J* = 1.9 Hz, 1H), 7.33 (t, *J* = 1.8 Hz, 1H), 7.08–7.03 (m, 2H), 6.69–6.65 (m, 2H), 3.47 (s, 2H). ESI-MS: *m*/*z* = 338.34 [M + H]^+^.

#### 4.2.11. N-(2-bromophenyl)-2-(4-hydroxyphenyl)acetamide (**12k**)

The title compound was obtained similarly to **12a**. 3,5-dimethylaniline **10a** was replaced with 2-bromoaniline **10g.** A brown solid was obtained; yield: 78.4%. ^1^H NMR (400 MHz, DMSO-*d6*) δ 9.40 (s, 1H), 9.25 (s, 1H), 7.59 (ddd, *J* = 7.9, 4.6, 1.4 Hz, 2H), 7.12 (d, *J* = 8.4 Hz, 2H), 7.09–6.97 (m, 2H), 6.70–6.66 (m, 2H), 3.54 (s, 2H). ESI-MS: *m*/*z* = 306.01 [M + H]^+^.

#### 4.2.12. N-(2-bromo-3-methylphenyl)-2-(4-hydroxyphenyl)acetamide (**12l**)

The title compound was obtained similarly to **12b.** 3,5-dimethylaniline **10a** was replaced with 3-bromo-2-methylaniline **10h.** A brown solid was obtained; yield: 16.9%. ^1^H NMR (400 MHz, DMSO-*d6*) δ 9.65 (s, 1H), 9.24 (s, 1H), 7.39 (d, *J* = 8.1 Hz, 1H), 7.27 (d, *J* = 7.9 Hz, 1H), 7.07 (dd, *J* = 16.9, 8.2 Hz, 3H), 6.68–6.64 (m, 2H), 3.48 (s, 2H), 2.13 (d, *J* = 12.4 Hz, 3H). ESI-MS: *m*/*z* = 320.02 [M + H]^+^.

#### 4.2.13. 2-(4-hydroxyphenyl)-N-(4-(methylthio)phenyl)acetamide (**12m**)

The title compound was obtained similarly to **12b.** 3,5-dimethylaniline **10a** was replaced with 4-(methylthio)aniline **10i.** A white solid was obtained; yield: 41.7%. ^1^H NMR (600 MHz, DMSO-*d6*) δ 10.05 (s, 1H), 9.24 (s, 1H), 7.57–7.52 (m, 2H), 7.24–7.18 (m, 2H), 7.11 (d, *J* = 8.5 Hz, 2H), 6.74–6.66 (m, 2H), 3.48 (s, 2H), 2.43 (s, 3H). ESI-MS: *m*/*z* = 274.08 [M + H]^+^.

#### 4.2.14. 2-(4-hydroxyphenyl)-N-(3-(methylthio)phenyl)acetamide (**12m-1**)

The title compound was obtained similarly to **12b.** 3,5-dimethylaniline **10a** was replaced with 3-(methylthio)aniline **10j.** A white solid was obtained; yield: 63.6%. ^1^H NMR (400 MHz, DMSO-*d6*) δ 10.05 (d, *J* = 14.8 Hz, 1H), 9.24 (s, 1H), 7.56–7.50 (m, 1H), 7.28 (ddd, *J* = 8.1, 2.0, 1.0 Hz, 1H), 7.21–7.15 (m, 1H), 7.08–7.03 (m, 2H), 6.87 (ddd, *J* = 7.8, 1.9, 1.0 Hz, 1H), 6.68–6.63 (m, 2H), 3.44 (s, 2H), 2.39 (s, 3H). ESI-MS: *m*/*z* = 274.09 [M + H]^+^.

#### 4.2.15. N-(4-ethynylphenyl)-2-(4-hydroxyphenyl)acetamide (**12n**)

The title compound was obtained similarly to **12b.** 3,5-dimethylaniline **10a** was replaced with 4-ethynylaniline **10k.** A white solid was obtained; yield: 55.3%. ^1^H NMR (600 MHz, DMSO-*d6*) δ 10.23 (s, 1H), 9.25 (s, 1H), 7.61 (d, *J* = 8.8 Hz, 2H), 7.41–7.38 (m, 2H), 7.11 (d, *J* = 8.5 Hz, 2H), 6.72–6.68 (m, 2H), 3.50 (s, 2H), 2.38 (d, *J* = 5.5 Hz, 1H). ESI-MS: *m*/*z* = 252.09 [M + H]^+^.

#### 4.2.16. N-(3-ethynylphenyl)-2-(4-hydroxyphenyl)acetamide (**12n-1**)

The title compound was obtained similarly to **12b.** 3,5-dimethylaniline **10a** was replaced with 3-ethynylaniline **10l.** A white solid was obtained; yield: 63.1%. ^1^H NMR (600 MHz, DMSO-*d6*) δ 10.16 (s, 1H), 9.25 (s, 1H), 7.78 (t, *J* = 1.7 Hz, 1H), 7.58–7.53 (m, 1H), 7.31 (td, *J* = 7.9, 4.5 Hz, 1H), 7.15–7.10 (m, 3H), 6.72–6.69 (m, 2H), 4.15 (s, 1H), 3.50 (s, 2H). ESI-MS: *m*/*z* = 252.09 [M + H]^+^.

#### 4.2.17. N-([1,1′-biphenyl]-4-yl)-2-(4-hydroxyphenyl)acetamide (**12o**)

The title compound was obtained similarly to **12b.** 3,5-dimethylaniline **10a** was replaced with [1,1′-biphenyl]-4-amine **10m.** A white solid was obtained; yield: 66.0%. ^1^H NMR (600 MHz, DMSO-*d6*) δ 10.16 (s, 1H), 9.25 (s, 1H), 7.69 (d, *J* = 8.7 Hz, 2H), 7.64–7.59 (m, 4H), 7.43 (dd, *J* = 10.6, 5.0 Hz, 2H), 7.34–7.30 (m, 1H), 7.13 (d, *J* = 8.5 Hz, 2H), 6.73–6.69 (m, 2H), 3.52 (s, 2H). ESI-MS: *m*/*z* = 304.12 [M + H]^+^.

#### 4.2.18. N-([1,1′-biphenyl]-3-yl)-2-(4-hydroxyphenyl)acetamide (**12p**)

The title compound was obtained similarly to **12b.** 3,5-dimethylaniline **10a** was replaced with [1,1′-biphenyl]-3-amine **10n.** A white solid was obtained; yield: 67.3%. ^1^H NMR (600 MHz, DMSO-*d6*) δ 10.16 (s, 1H), 9.25 (s, 1H), 7.92 (t, *J* = 1.8 Hz, 1H), 7.62–7.55 (m, 3H), 7.50–7.44 (m, 2H), 7.41–7.34 (m, 2H), 7.34–7.27 (m, 1H), 7.14 (d, *J* = 8.5 Hz, 2H), 6.77–6.67 (m, 2H), 3.52 (s, 2H). ESI-MS: *m*/*z* = 304.12 [M + H]^+^.

#### 4.2.19. N-(4-cyanophenyl)-2-(4-hydroxyphenyl)acetamide (**12q**)

The title compound was obtained similarly to **12a**. 3,5-dimethylaniline **10a** was replaced with 4-aminobenzonitrile **10o.** A white solid was obtained; yield: 46.6%. ^1^H NMR (400 MHz, DMSO-*d6*) δ 10.49 (s, 1H), 9.26 (s, 1H), 7.74–7.71 (m, 4H), 7.09–7.05 (m, 2H), 6.67–6.65 (m, 2H), 3.50 (s, 2H). ESI-MS: *m*/*z* = 253.09 [M + H]^+^.

#### 4.2.20. N-(4-(diethylamino)phenyl)-2-(4-hydroxyphenyl)acetamide (**12r**)

The title compound was obtained similarly to **12b.** 3,5-dimethylaniline **10a** was replaced with N1,N1-diethylbenzene-1,4-diamine **10p**. A brown solid was obtained; yield: 43.6%. ^1^H NMR (400 MHz, METHANOL-*d3*) δ 7.32–7.27 (m, 2H), 7.16–7.12 (m, 2H), 6.74–6.70 (m, 3H), 6.68–6.66 (m, 1H), 3.50 (s, 2H), 3.30–3.27 (m, 4H), 1.09 (t, *J* = 7.1 Hz, 6H). ESI-MS: *m*/*z* = 299.17 [M + H]^+^.

#### 4.2.21. 2-(4-hydroxyphenyl)-N-(4-methoxyphenyl)acetamide (**12s**)

The title compound was obtained similarly to **12b.** 3,5-dimethylaniline **10a** was replaced with 4-methoxyaniline **10q.** A white solid was obtained; yield: 46.7%.^1^H NMR (400 MHz, DMSO-*d6*) δ 9.88 (s, 1H), 9.21 (s, 1H), 7.47–7.43 (m, 2H), 7.09–7.05 (m, 2H), 6.84–6.80 (m, 2H), 6.68–6.64 (m, 2H), 3.67 (s, 3H), 3.41 (s, 2H). ESI-MS: *m*/*z* = 258.10 [M + H]^+^.

#### 4.2.22. (E)-3-(4-hydroxyphenyl)-N-(3-(methylthio)phenyl)acrylamide (**12t**)

The title compound was obtained similarly to **12b.** 3,5-dimethylaniline **10a** was replaced with 3-(methylthio)aniline **10r.** 4-hydroxybenzoic acid **11b** was replaced with (E)-3-(4-hydroxyphenyl)acrylic acid **11f.** A pale yellow solid was obtained; yield: 17.5%. ^1^H NMR (600 MHz, DMSO-*d6*) δ 10.09 (s, 1H), 9.92 (s, 1H), 7.69 (t, *J* = 1.8 Hz, 1H), 7.52–7.50 (m, 1H), 7.46 (dd, *J* = 9.0, 2.2 Hz, 2H), 7.41 (dd, *J* = 8.1, 1.1 Hz, 1H), 7.26 (t, *J* = 8.0 Hz, 1H), 6.94 (ddd, *J* = 7.8, 1.7, 0.8 Hz, 1H), 6.85–6.81 (m, 2H), 6.59 (d, *J* = 15.6 Hz, 1H), 2.47 (s, 3H). ESI-MS: *m*/*z* = 286.08 [M + H]^+^.

#### 4.2.23. (E)-N-(4-bromophenyl)-3-(4-hydroxyphenyl)acrylamide (**12u**)

The title compound was obtained similarly to **12t.** 3,5-dimethylaniline **10a** was replaced with 4-bromoaniline **10s.** A pale yellow solid was obtained; yield: 19.7%. ^1^H NMR (600 MHz, DMSO-*d6*) δ 10.20 (s, 1H), 9.93 (s, 1H), 7.67–7.65 (m, 2H), 7.50 (dt, *J* = 4.7, 3.4 Hz, 3H), 7.48–7.45 (m, 2H), 6.84–6.81 (m, 2H), 6.58 (d, *J* = 15.6 Hz, 1H). ESI-MS: *m*/*z* = 318.00 [M + H]^+^.

#### 4.2.24. (E)-3-(4-hydroxyphenyl)-N-(3-iodophenyl)acrylamide (**12v**)

The title compound was obtained similarly to **12t.** 3,5-dimethylaniline **10a** was replaced with 3-iodoaniline **10e.** A pale yellow solid was obtained; yield: 18.3%. ^1^H NMR (600 MHz, DMSO-*d6*) δ 10.17 (s, 1H), 9.94 (s, 1H), 8.21 (t, *J* = 1.8 Hz, 1H), 7.62–7.59 (m, 1H), 7.48 (dd, *J* = 20.4, 12.1 Hz, 3H), 7.40 (ddd, *J* = 7.8, 1.6, 0.9 Hz, 1H), 7.12 (t, *J* = 8.0 Hz, 1H), 6.84–6.80 (m, 2H), 6.57 (d, *J* = 15.6 Hz, 1H). ESI-MS: *m*/*z* = 365.98 [M + H]^+^.

#### 4.2.25. (E)-N-(3-bromo-5-chlorophenyl)-3-(4-hydroxyphenyl)acrylamide (**12w**)

The title compound was obtained similarly to **12t.** 3,5-dimethylaniline **10a** was replaced with 3-bromo-5-chloroaniline **10f.** A pale yellow solid was obtained; yield: 19.2%. ^1^H NMR (600 MHz, DMSO-*d6*) δ 10.40 (d, *J* = 10.9 Hz, 1H), 9.99 (d, *J* = 12.6 Hz, 1H), 7.90–7.86 (m, 1H), 7.82–7.78 (m, 1H), 7.54 (t, *J* = 9.2 Hz, 1H), 7.48 (d, *J* = 8.6 Hz, 2H), 7.38 (t, *J* = 1.8 Hz, 1H), 6.86–6.81 (m, 2H), 6.53 (d, *J* = 15.6 Hz, 1H). ESI-MS: *m*/*z* = 374.24 [M + Na]+.

#### 4.2.26. (E)-N-(3,5-dimethylphenyl)-3-(4-hydroxy-3-methoxyphenyl)acrylamide (**12x**)

The title compound was obtained similarly to **12t.** 4-hydroxybenzoic acid **11b** was replaced with (E)-3-(4-hydroxy-3-methoxyphenyl)acrylic acid **11g.** A pale yellow solid was obtained; yield: 61.5%. ^1^H NMR (400 MHz, DMSO-*d6*) δ 10.03 (d, *J* = 15.4 Hz, 1H), 8.79 (s, 1H), 6.83–6.08 (m, 6H), 4.21 (d, *J* = 6.4 Hz, 1H), 3.65 (s, 3H), 2.85 (dd, *J* = 16.0, 7.2 Hz, 1H), 2.09 (d, *J* = 70.2 Hz, 6H). ESI-MS: *m*/*z* = 298.14 [M + H]^+^.

#### 4.2.27. 2-(3,4-dihydroxyphenyl)-N-(3,5-dimethylphenyl)acetamide (**12y**)

The title compound was obtained similarly to **12b.** 4-hydroxybenzoic acid **11b** was replaced with 2-(3,4-dihydroxyphenyl)acetic acid **11h.** A white solid was obtained; yield: 8.9%. ^1^H NMR (400 MHz, DMSO-*d6*) δ 9.84 (s, 1H), 8.80 (s, 1H), 8.68 (s, 1H), 7.16 (t, *J* = 3.4 Hz, 2H), 6.68 (d, *J* = 2.1 Hz, 1H), 6.61 (dd, *J* = 9.2, 4.4 Hz, 2H), 6.50 (dd, *J* = 8.0, 2.1 Hz, 1H), 3.36–3.32 (m, 2H), 2.19–2.11 (m, 6H). ESI-MS: *m*/*z* = 272.12 [M + H]^+^.

### 4.3. Procedure for the Syntheses of Compounds ***15a–15d***

#### 4.3.1. Ethyl 4-(4-(2-((3,5-dimethylphenyl)amino)-2-oxoethyl)phenoxy)butanoate (**15a**)

To a stirring solution of 3,5-dimethylaniline **12a** (0.51 g, 2 mmol) in 2-butanone (8 mL) were added Cs_2_CO_3_ (1.95 g, 6 mmol) and methyl ethyl 4-bromobutanoate **14a** (0.61 g, 3.4 mmol). The reaction mixture was heated to 50℃ for 12 h. Upon completion, the reaction mixture was diluted with EA (16 mL) and washed with brine (20 mL). The organic phase was dried over anhydrous sodium sulfate, filtered, and evaporated under reduced pressure. The crude product was purified by flash chromatography using PE/EA (5:1 to 3:1) as an eluent to afford **15a** as a brown solid (0.48g, 64.5%).^1^H NMR (400 MHz, DMSO-*d6*) δ 9.91 (s, 1H), 7.18 (dd, *J* = 6.9, 4.8 Hz, 4H), 6.85–6.79 (m, 2H), 6.64–6.59 (m, 1H), 4.05–3.98 (m, 2H), 3.90 (t, *J* = 6.4 Hz, 2H), 3.46 (s, 2H), 2.43–2.38 (m, 2H), 2.18 (t, *J* = 7.8 Hz, 6H), 1.96–1.86 (m, 2H), 1.16–1.09 (m, 3H). ^13^C NMR (151 MHz, DMSO-*d6*) δ 173.02, 169.79, 157.66, 139.56, 138.09, 130.49, 128.53, 125.15, 117.34, 114.73, 66.89, 60.29, 42.97, 30.61, 21.54, 14.55. HRMS (ESI) *m*/*z* [M+H] + calculated for C_22_H_27_NO_4_: 370.2018; found: 370.2103.

#### 4.3.2. Methyl2-(4-(2-((3,5-dimethylphenyl)amino)-2-oxoethyl)phenoxy)-2-phenylacetate (**15b**)

The title compound was obtained similarly to **15a.** Methyl 2-(bromomethyl)acrylate **14a** was replaced with ethyl methyl 2-bromo-2-phenylacetate **14b.** A white solid was obtained; yield: 94.3%. ^1^H NMR (600 MHz, DMSO-*d6*) δ 9.93 (s, 1H), 7.55–7.52 (m, 2H), 7.44–7.36 (m, 3H), 7.24–7.17 (m, 4H), 6.94–6.91 (m, 2H), 6.67 (s, 1H), 5.98 (s, 1H), 3.65 (s, 3H), 3.51 (s, 2H), 2.54 (s, 1H), 2.21 (s, 6H). ^13^C NMR (151 MHz, DMSO-*d6*) δ 171.63, 169.77, 156.57, 139.53, 138.09, 137.47, 130.43, 128.90, 128.82, 128.72, 127.77, 125.17, 117.39, 115.51, 78.51, 52.89, 42.93, 21.54. HRMS (ESI) *m*/*z* [M+H] + calculated for C_24_H_23_NO_4_: 390.1705; found: 390.1700.

#### 4.3.3. Methyl2-(4-(2-((3,5-dimethylphenyl)amino)-2-oxoethyl)phenoxy)-2methylpropanoate (**15c**)

The title compound was obtained similarly to **15a.** Methyl 2-(bromomethyl)acrylate **14a** was replaced with methyl 2-bromo-2-methylpropanoate **14c.** A yellow solid was obtained; yield: 90.1%. ^1^H NMR (400 MHz, DMSO-*d6*) δ 9.94 (s, 1H), 7.18–7.14 (m, 4H), 6.71–6.67 (m, 2H), 6.64–6.62 (m, 1H), 3.65 (d, *J* = 1.8 Hz, 3H), 3.48 (s, 2H), 2.17 (d, *J* = 0.5 Hz, 6H), 1.46 (s, 6H). ^13^C NMR (151 MHz, DMSO-*d6*) δ 174.28, 169.65, 154.10, 139.48, 138.13, 130.40, 130.13, 125.22, 119.17, 117.35, 79.05, 52.83, 42.91, 25.44, 21.53. HRMS (ESI) *m*/*z* [M + H] + calculated for C_21_H_25_NO_4_: 356.1862; found: 356.1856.

### 4.4. Procedure for Syntheses of Compounds ***16a–b***

#### 4.4.1. 4-(4-(2-((3,5-dimethylphenyl)amino)-2-oxoethyl)phenoxy)butanoic acid (**16a**)

To a stirring solution of methyl2-((4-(2-((3,5-dimethylphenyl)amino)-2-oxoethyl)phenoxy)methyl)acrylate **15a** (0.37 g, 1 mmol) in THF (2 mL) were added LiOH.H_2_O (84 mg, 2 mmol) and H2O (2 mL) under ice bath. After being stirred for 6 h, the reaction mixture was diluted with EA (5 mL) and acidified with 1N HCl. The organic phase was washed with brine (3 × 5 mL), dried over anhydrous Na_2_SO_4_ and concentrated. The residue was diluted with EA (1 mL) and PE (4 mL) under ice bath. The precipitate was collected by filtration and dried over vacuum oven. A white solid was obtained (0.29g, 85.0%).^1^H NMR (400 MHz, DMSO-*d6*) δ 12.12 (s, 1H), 9.91 (s, 1H), 7.20–7.15 (m, 4H), 6.86–6.80 (m, 2H), 6.63 (d, *J* = 0.7 Hz, 1H), 3.90 (t, *J* = 6.4 Hz, 2H), 3.47 (s, 2H), 2.33 (t, *J* = 7.3 Hz, 2H), 2.17 (d, *J* = 0.5 Hz, 6H), 1.92–1.82 (m, 2H). ^13^C NMR (151 MHz, DMSO-*d6*) δ 174.58, 169.79, 157.72, 139.57, 138.09, 130.49, 128.50, 125.15, 117.35, 114.75, 67.00, 42.98, 30.57, 21.55. HRMS (ESI) *m*/*z* [M + H] + calculated for C_20_H_23_NO_4_: 342.1705; found: 342.1700.

#### 4.4.2. 4-(4-(2-((3,5-dimethylphenyl)amino)-2-oxoethyl)phenoxy)butanoic acid (**16b**)

The title compound was obtained similarly to **16a.** Methyl2-((4-(2-((3,5-dimethylphenyl)amino)-2-oxoethyl)phenoxy)methyl)acrylate **15a** was replaced with methyl2-(4-(2-((3,5-dimethylphenyl)amino)-2-oxoethyl)phenoxy)-2-phenylacetate **15b**. A white solid was obtained; yield: 80.0%.^1^H NMR (600 MHz, DMSO) δ 9.94 (s, 1H), 7.52 (d, *J* = 7.4 Hz, 1H), 7.34 (dt, *J* = 26.6, 7.2 Hz, 1H), 7.20 (d, *J* = 6.5 Hz, 1H), 6.89 (d, *J* = 8.5 Hz, 2H), 6.66 (s, 1H), 5.66 (s, 1H), 3.50 (s, 2H), 2.21 (s, 6H). ^13^C NMR (151 MHz, DMSO-*d6*) δ 171.63, 169.77, 156.57, 139.53, 138.09, 137.47, 130.43, 128.90, 128.82, 128.72, 127.77, 125.17, 117.39, 115.51, 78.51, 42.93, 21.54. HRMS (ESI) *m*/*z* [M + H] + calculated for C_24_H_23_NO_4_: 390.1705; found: 390.1700.

### 4.5. Procedure for the Syntheses of Intermediates ***17a–17g***

#### 4.5.1. 4-(3-((3,5-dimethylphenyl)amino)propyl)phenol (**17a**)

To a stirring solution of LiAlH_4_ (0.42g, 12 mmol) in THF (15 mL) was added N-(3,5-dimethylphenyl)-3-(4-hydroxyphenyl)propanamide **12c** (1.28 g, 5 mmol) portion-wise. The reaction mixture was heated to reflux for 12 h. Upon completion, the reaction mixture was diluted with ether (16 mL) and cooled down to 0 ℃. The solution was further processed with water (0.42 mL), 15% sodium hydroxide (1.26 mL), water (0.42 mL), and anhydrous magnesium sulfate (2 g) successively, and the insoluble matter was removed by filtration. The organic phase was dried over anhydrous magnesium sulfate, filtered, and evaporated under reduced pressure. The crude product was purified by flash chromatography using hexane: EA(6:1 to 3:1) as eluent to afford **17a** as a white solid (0.95 g, 69.2%). ^1^H NMR (400 MHz, DMSO-*d6*) δ 9.13 (s, 1H), 7.02 (d, *J* = 8.4 Hz, 2H), 6.69 (d, *J* = 8.4 Hz, 2H), 6.16 (d, *J* = 6.7 Hz, 3H), 5.36 (t, *J* = 5.4 Hz, 1H), 2.95 (dd, *J* = 12.8, 6.6 Hz, 2H), 2.56 (dd, *J* = 12.9, 5.3 Hz, 2H), 2.14 (s, 6H), 1.82–1.73 (m, 2H). ESI-MS: *m*/*z* = 256.15 [M + H]^+^.

#### 4.5.2. 4-(2-((3-bromo-5-chlorophenyl)amino)ethyl)phenol (**17b**)

The title compound was obtained similarly to **17a.** N-(3,5-dimethylphenyl)-3-(4-hydroxyphenyl)propanamide **12c** was replaced with N-(3-bromo-5-chlorophenyl)-2-(4-hydroxyphenyl)acetamide **12j.** A white solid was obtained; yield: 59.3%. ^1^H NMR (400 MHz, DMSO-*d6*) δ 9.18 (d, *J* = 2.8 Hz, 1H), 7.06 (d, *J* = 8.3 Hz, 2H), 6.77–6.64 (m, 4H), 6.59 (dd, *J* = 4.6, 2.6 Hz, 1H), 6.42 (dt, *J* = 10.6, 6.4 Hz, 1H), 3.20 (dd, *J* = 12.9, 7.3 Hz, 2H), 2.71 (dd, *J* = 13.1, 5.9 Hz, 2H). ESI-MS: *m*/*z* = 327.96 [M + H]^+^.

#### 4.5.3. 4-(2-((3-(methylthio)phenyl)amino)ethyl)phenol **(17c**)

The title compound was obtained similarly to **17a.** N-(3,5-dimethylphenyl)-3-(4-hydroxyphenyl)propanamide **12c** was replaced with 2-(4-hydroxyphenyl)-N-(3-(methylthio)phenyl)acetamide **12m-1.** A white solid was obtained; yield: 61.8%. ^1^H NMR (600 MHz, DMSO-*d6*) δ 9.17 (s, 1H), 7.07–7.03 (m, 2H), 7.00 (t, *J* = 7.9 Hz, 1H), 6.70–6.66 (m, 2H), 6.44 (t, *J* = 2.0 Hz, 1H), 6.41 (ddd, *J* = 7.7, 1.7, 0.8 Hz, 1H), 6.36 (ddd, *J* = 8.1, 2.2, 0.7 Hz, 1H), 5.67 (t, *J* = 5.6 Hz, 1H), 3.15 (dt, *J* = 7.6, 5.8 Hz, 2H), 2.72–2.68 (m, 2H), 2.39 (s, 3H). ESI-MS: *m*/*z* = 260.08 [M + H]^+^.

#### 4.5.4. 4-(2-((3-ethynylphenyl)amino)ethyl)phenol (**17d**)

The title compound was obtained similarly to **17a.** N-(3,5-dimethylphenyl)-3-(4-hydroxyphenyl)propanamide **12c** was replaced with N-(3-ethynylphenyl)-2-(4-hydroxyphenyl)acetamide **12n-1.** A white solid was obtained; yield: 57.8%. ^1^H NMR (600 MHz, DMSO-*d6*) δ 9.16 (s, 1H), 7.08–7.03 (m, 3H), 6.70–6.66 (m, 2H), 6.65–6.59 (m, 3H), 5.80 (t, *J* = 5.6 Hz, 1H), 3.99 (s, 1H), 3.16 (dt, *J* = 7.6, 5.8 Hz, 2H), 2.72–2.67 (m, 2H).ESI-MS: *m*/*z* = 238.09 [M + H]^+^.

#### 4.5.5. 4-(2-([1,1′-biphenyl]-4-ylamino)ethyl)phenol (**17e**)

The title compound was obtained similarly to **17a.** N-(3,5-dimethylphenyl)-3-(4-hydroxyphenyl)propanamide **12c** was replaced with N-([1,1′-biphenyl]-3-yl)-2-(4-hydroxyphenyl)acetamide **12p.** A white solid was obtained; yield: 62.8%. ^1^H NMR (600 MHz, DMSO-*d6*) δ 9.17 (s, 1H), 7.57 (dt, *J* = 8.1, 1.5 Hz, 2H), 7.46–7.40 (m, 2H), 7.35–7.30 (m, 1H), 7.16 (t, *J* = 7.8 Hz, 1H), 7.10–7.05 (m, 2H), 6.80 (ddd, *J* = 2.9, 2.5, 1.3 Hz, 2H), 6.71–6.67 (m, 2H), 6.59 (ddd, *J* = 8.0, 2.1, 0.7 Hz, 1H), 5.71 (t, *J* = 5.6 Hz, 1H), 3.23 (dt, *J* = 7.6, 5.9 Hz, 2H), 2.77–2.72 (m, 2H). *m*/*z* = 290.12 [M + H]^+^.

#### 4.5.6. 4-(2-(naphthalen-2-ylamino)ethyl)phenol (**17f**)

The title compound was obtained similarly to **17a.** N-(3,5-dimethylphenyl)-3-(4-hydroxyphenyl)propanamide **12c** was replaced with 2-(4-hydroxyphenyl)-N-(naphthalen-2-yl)acetamide **22e.** A white solid was obtained; yield: 30.7%.

^1^H NMR (400 MHz, DMSO-*d6*) δ 9.19 (s, 1H), 7.67–7.56 (m, 3H), 7.31 (dd, *J* = 11.0, 4.0 Hz, 1H), 7.11 (dd, *J* = 11.7, 5.1 Hz, 3H), 6.99 (dd, *J* = 8.9, 2.2 Hz, 1H), 6.73 (dd, *J* = 9.9, 5.2 Hz, 3H), 5.96 (t, *J* = 5.4 Hz, 1H), 3.29 (dd, *J* = 13.3, 7.3 Hz, 2H), 2.82 (t, *J* = 7.5 Hz, 2H). ESI-MS: *m*/*z* = 264.11 [M + H]^+^.

#### 4.5.7. 4-(2-((3,5-dimethylphenyl)amino)ethyl)phenol (**17g**)

The title compound was obtained similarly to **17a.** N-(3,5-dimethylphenyl)-3-(4-hydroxyphenyl) propanamide **12c** was replaced with N-(3,5-dimethylphenyl)-2-(4-hydroxyphenyl)acetamide **12a.** A white solid was obtained; yield: 47.3%. ^1^H NMR (400 MHz, DMSO-*d6*) δ 9.13 (s, 1H), 7.06–6.98 (m, 2H), 6.69–6.62 (m, 2H), 6.18–6.09 (m, 3H), 5.33 (t, *J* = 5.6 Hz, 1H), 3.12–3.05 (m, 2H), 2.65 (dd, *J* = 9.2, 5.8 Hz, 2H), 2.08 (t, *J* = 7.0 Hz, 6H). ESI-MS: *m*/*z* = 242.15 [M + H]^+^.

### 4.6. Procedure for the Syntheses of ***18a–18g***

#### 4.6.1. 2-(4-(3-((3,5-dimethylphenyl)amino)propyl)phenoxy)-2-methylpropanoic acid (**18a**)

To a stirred solution of 4-(3-((3,5-dimethylphenyl)amino)propyl)phenol **17a** (0.98 g, 3 mmol) in acetone (12 mL) was added NaOH (1.44 g, 36 mmol). The reaction mixture was cooled down to 0 ℃ for 0.5 h, and then chloroform (1.08 mg, 9 mmol) was added dropwise. Afterwards, the solution was stirred for another 6 h. After completion, the reaction mixture was diluted with EA(25 mL) and acidified with 1N HCl (pH < 3). The organic phase was washed with brine (3 × 25 mL), dried over anhydrous Na_2_SO_4_ and concentrated. The residue was diluted with EA (4 mL) and PE (20 mL) under ice bath. The precipitate was collected by filtration and dried over vacuum oven. A red solid was obtained (yield: 0.97 g, 76.3%). ^1^H NMR (600 MHz, DMSO-*d6*) δ 10.50 (s, 1H), 7.11–7.06 (m, 2H), 6.98 (dd, *J* = 24.0, 15.4 Hz,3H), 6.78–6.74 (m, 2H), 3.25–3.04 (m, 2H), 2.62–2.57 (m, 6H), 1.91–1.85 (m, 2H), 1.48 (s, 6H). ^13^C NMR (151 MHz, DMSO-*d6*) δ 175.52, 154.03, 139.75, 137.31, 134.33, 129.98, 129.42, 120.35, 119.06, 78.77, 50.31, 30.77, 27.88, 25.53, 21.29. HRMS (ESI) *m*/*z* [M + H] + calculated for C_21_H_27_NO_3_: 342.2069; found: 342.2064.

#### 4.6.2. 2-(4-(2-((3-bromo-5-chlorophenyl)amino)ethyl)phenoxy)-2-methylpropanoic acid (**18b**)

The title compound was obtained similarly to **18a.** 4-(3-((3,5-dimethylphenyl)amino)propyl)phenol **17a** was replaced with 4-(2-((3-bromo-5-chlorophenyl)amino)ethyl)phenol **17b.** A pale yellow solid was obtained; yield: 73.1%. ^1^H NMR (600 MHz, DMSO-*d6*) δ 7.14 (d, *J* = 8.0 Hz,2H), 6.75 (d, *J* = 8.3 Hz, 2H), 6.69 (dt, *J* = 6.7, 1.7 Hz, 1H), 6.58 (dt, *J* = 9.4, 2.0 Hz, 1H), 6.54–6.31 (m, 1H), 3.21 (dd, *J* = 12.8, 7.1 Hz, 2H), 2.73 (dd, *J* = 14.2, 6.8 Hz, 2H), 1.49–1.45 (m, 6H). ^13^C NMR (151 MHz, DMSO-*d6*) δ 175.71, 151.60, 150.68, 135.02, 130.84, 129.80, 123.10, 118.83, 117.14, 115.30, 113.35, 78.93, 44.57, 34.06, 25.68.HRMS (ESI) *m*/*z* [M + H] + calculated for C_18_H_19_NO_3_: 414.0294; found: 414.0288.

#### 4.6.3. 2-methyl-2-(4-(2-((3-(methylthio)phenyl)amino)ethyl)phenoxy)propanoic acid (**18c**)

The title compound was obtained similarly to **18a.** 4-(3-((3,5-dimethylphenyl)amino)propyl)phenol **17a** was replaced with 4-(2-((3-(methylthio)phenyl)amino)ethyl)phenol **17c.** A yellow oil was obtained; yield: 72.1%. ^1^H NMR (600 MHz, DMSO-*d6*) δ 12.89 (s, 1H), 7.19–7.13 (m, 2H), 7.00 (t, *J* = 7.9 Hz, 1H), 6.78–6.74 (m,2H), 6.45 (t, *J* = 2.0 Hz, 1H), 6.41 (ddd, *J* = 7.6, 1.6, 0.7 Hz, 1H), 6.37 (ddd, *J* = 8.1, 2.2, 0.7 Hz, 1H), 5.72 (s, 1H), 3.19 (t, *J* = 7.5 Hz,2H), 2.76–2.73 (m,2H), 2.40 (s, 3H), 1.52–1.48 (m, 6H). ^13^C NMR (151 MHz, DMSO-*d6*) δ 175.60, 154.08, 149.62, 138.79, 129.85, 129.83, 118.92, 113.73, 109.65, 109.41, 78.74, 45.02, 34.48, 25.48, 15.08. HRMS (ESI) *m*/*z* [M + H] + calculated for C_19_H_23_NO_3_S: 346.1477; found: 346.1470.

#### 4.6.4. 2-(4-(2-((3-ethynylphenyl)amino)ethyl)phenoxy)-2-methylpropanoic acid (**18d**)

The title compound was obtained similarly to **18a.** 4-(3-((3,5-dimethylphenyl)amino)propyl)phenol **17a** was replaced with 4-(2-((3-ethynylphenyl)amino)ethyl)phenol **17d.** A brown oil was obtained; yield: 71.3%. ^1^H NMR (600 MHz, DMSO-*d6*) δ 7.14 (d, *J* = 8.3 Hz, 2H), 7.06 (t, *J* = 7.8 Hz, 1H), 6.76 (d, *J* = 8.5 Hz, 2H), 6.66–6.59 (m, 3H), 5.85 (s, 1H), 3.99 (s, 1H), 3.19 (t, *J* = 7.4 Hz, 2H), 2.74 (t, *J* = 7.5 Hz, 2H), 1.48 (d, *J* = 9.5 Hz, 6H). ^13^C NMR (151 MHz, DMSO-*d6*) δ 170.83, 154.26, 149.17, 133.02, 129.77, 129.68, 122.56, 119.39, 118.87, 115.09, 113.31, 84.95, 79.67, 78.90, 44.92, 34.36, 25.56. HRMS (ESI) *m*/*z* [M + H] + calculated for C_20_H_21_NO_3_: 324.1599; found: 324.1594.

#### 4.6.5. 2-(4-(2-([1,1′-biphenyl]-4-ylamino)ethyl)phenoxy)-2-methylpropanoic acid (**18e**)

The title compound was obtained similarly to **18a.** 4-(3-((3,5-dimethylphenyl)amino)propyl)phenol **17a** was replaced with 4-(2-([1,1′-biphenyl]-4-ylamino)ethyl)phenol **17e.** A yellow solid was obtained; yield: 67.9%. ^1^H NMR (600 MHz, DMSO-*d6*) δ 12.93 (s, 1H), 7.59–7.56 (m, 2H), 7.45–7.40 (m, 2H), 7.34–7.30 (m, 1H), 7.20–7.13 (m, 3H), 6.80 (ddd, *J* = 13.1, 5.1, 3.7 Hz, 2H), 6.78–6.74 (m, 2H), 6.61–6.58 (m, 1H), 5.76 (s, 1H), 3.27 (t, *J* = 7.5 Hz, 2H), 2.80 (t, *J* = 7.5 Hz, 2H), 1.48 (s, 6H). ^13^C NMR (151 MHz, DMSO-*d6*) δ 175.61, 154.09, 149.64, 141.58, 133.45, 129.89, 129.21, 127.58, 127.05, 118.95, 114.82, 111.81, 110.76, 78.75, 45.23, 34.57, 25.53. HRMS (ESI) *m*/*z* [M + H] + calculated for C_24_H_25_NO_3_: 376.1912; found: 376.1907.

#### 4.6.6. 2-methyl-2-(4-(2-(naphthalen-2-ylamino)ethyl)phenoxy)propanoic acid (**18f**)

The title compound was obtained similarly to **18a.** 4-(3-((3,5-dimethylphenyl)amino)propyl)phenol **17a** was replaced with 4-(2-(naphthalen-2-ylamino)ethyl)phenol **17f.** A white solid was obtained; yield: 69.9%. ^1^H NMR (600 MHz, DMSO-*d6*) δ 7.62 (d, *J* = 8.0 Hz, 1H), 7.58 (dd, *J* = 8.4, 4.0 Hz, 2H), 7.28 (ddd, *J* = 8.1, 6.8, 1.2 Hz, 1H), 7.20 (d, *J* = 8.3 Hz, 2H), 7.09 (ddd, *J* = 8.0, 6.8, 1.1 Hz, 1H), 7.00–6.96 (m,2H), 6.78 (d, *J* = 8.4 Hz, 1H), 6.74 (d, *J* = 2.0 Hz, 1H), 5.99 (s, 1H), 3.33–3.27 (m, 2H), 2.89–2.80 (m, 2H), 1.49 (d, *J* = 7.4 Hz, 6H). ^13^C NMR (151 MHz, DMSO-*d6*) δ 175.73, 154.28, 147.02, 135.71, 133.20, 129.84, 128.76, 127.85, 126.88, 126.39, 125.94, 121.40, 118.90, 118.78, 102.77, 78.85, 45.33, 34.28, 25.66. HRMS (ESI) *m*/*z* [M + H] + calculated for C_22_H_23_NO_3_: 350.1756; found: 350.1751.

#### 4.6.7. 2-(4-(2-((3,5-dimethylphenyl)amino)ethyl)phenoxy)-2-methylpropanoic acid (**18g**)

The title compound was obtained similarly to **18a.** 4-(3-((3,5-dimethylphenyl)amino)propyl)phenol **17a** was replaced with 4-(2-((3,5-dimethylphenyl)amino)ethyl)phenol **17g.** A pale brown solid was obtained; yield: 75.3%. ^1^H NMR (400 MHz, DMSO-*d6*) δ 7.12 (dd, *J* = 9.2, 2.5 Hz, 2H), 6.74–6.70 (m, 2H), 6.63 (dd, *J* = 21.5, 7.7 Hz, 3H), 3.28 (s, 2H), 2.80–2.74 (m, 2H), 2.18 (s, 6H), 1.46 (d, *J* = 11.8 Hz, 6H). ^13^C NMR (151 MHz, DMSO-*d6*) δ 175.49, 154.51, 139.60, 137.74, 130.76, 129.90, 129.47, 120.01, 119.02, 78.78, 51.06, 31.33, 25.50, 21.29. HRMS (ESI) *m*/*z* [M + H] + calculated for C_20_H_25_NO_3_: 328.1912; found: 328.1907.

### 4.7. Procedure for the Syntheses of ***19a–19x***

#### 4.7.1. 2-(4-(2-((3,5-dimethylphenyl)amino)-2-oxoethyl)phenoxy)-2-methylpropanoic acid (**19a**)

To a stirred solution of N-(3,5-dimethylphenyl)-2-(4-hydroxyphenyl)acetamide **12a** (0.51 g, 2 mmol) in acetone (6 mL) was added NaOH (0.96 g, 24 mmol). The reaction mixture was cooled down to 0 ℃ for 0.5 h, and then chloroform (0.72 mg, 6 mmol) was added dropwise. Afterwards, the solution was stirred for another 6 h. After completion, the reaction mixture was diluted with EA (15 mL) and acidified with 1N HCl (pH < 3). The organic phase was washed with brine (3 × 15 mL), dried over anhydrous Na_2_SO_4_ and concentrated. The residue was diluted with EA (2 mL) and PE (8 mL) under ice bath. The precipitate was collected by filtration and dried over vacuum oven. A white solid was obtained (0.48 g, 70.0%). ^1^H NMR (400 MHz, DMSO-*d6*) δ 10.03 (s, 1H), 7.19 (s, 2H), 7.07 (d, *J* = 8.0 Hz, 2H), 6.72 (d, *J* = 8.1 Hz, 2H), 6.61 (d, *J* = 7.4 Hz, 1H), 3.44 (s, 2H), 2.17 (s, 6H), 1.36 (s, 6H). ^13^C NMR (151 MHz, DMSO-*d6*) δ 175.51, 169.63, 154.47, 139.56, 138.09, 130.26, 128.67, 125.16, 118.75, 117.35, 78.79, 42.95, 25.53, 21.53. HRMS (ESI) *m*/*z* [M + H] + calculated for C_20_H_23_NO_4_: 342.1705; found: 342.1700.

#### 4.7.2. 2-(4-((3,5-dimethylphenyl) carbamoyl) phenoxy)-2-methylpropanoic acid (**19b**)

The title compound was obtained similarly to **19a.** N-(3,5-dimethylphenyl)-2-(4-hydroxyphenyl)acetamide **12a** was replaced with N-(3,5-dimethylphenyl)-4-hydroxybenzamide **12b** (0.53 g, 74.0%). ^1^H NMR (600 MHz, DMSO-*d6*) δ 13.17 (s, 1H), 9.92 (s, 1H), 7.93–7.76 (m, 2H), 7.38 (s, 2H), 6.95–6.83 (m, 2H), 6.73 (s, 1H), 2.26 (s, 6H), 1.57 (s, 6H). ^13^C NMR (151 MHz, DMSO-*d6*) δ 175.10, 165.28, 158.55, 139.57, 137.95, 129.62, 118.53, 117.54, 79.14, 25.54, 21.59. HRMS (ESI) *m*/*z* [M + H] + calculated for C_19_H_21_NO_4_: 328.1549; found: 328.1543.

#### 4.7.3. 2-(4-(3-((3,5-dimethylphenyl)amino)-3-oxopropyl)phenoxy)-2-methylpropanoic acid (**19c**)

The title compound was obtained similarly to **19a.** N-(3,5-dimethylphenyl)-2-(4-hydroxyphenyl)acetamide **12a** was replaced with N-(3,5-dimethylphenyl)-3-(4-hydroxyphenyl)propanamide **12c.** A white solid was obtained; yield: 72.8%. ^1^H NMR (400 MHz, DMSO-*d6*) δ 12.95 (s, 1H), 9.69 (s, 1H), 7.16 (s, 2H), 7.09 (d, *J* = 8.6 Hz, 2H), 6.73–6.69 (m, 2H), 6.63 (s, 1H), 2.78 (t, *J* = 7.6 Hz, 2H), 2.52 (t, *J* = 7.7 Hz, 2H), 2.18 (s, 6H), 1.44 (s, 6H). ^13^C NMR (151 MHz, DMSO-*d6*) δ 175.60, 170.83, 153.93, 139.50, 138.07, 134.80, 129.41, 125.06, 118.99, 117.34, 78.75, 38.53, 30.48, 25.50, 21.55. HRMS (ESI) *m*/*z* [M + H] + calculated for C_21_H_25_NO_4_: 356.1862; found: 356.1856.

#### 4.7.4. 2-(4-(4-((3,5-dimethylphenyl)amino)-4-oxobutyl)phenoxy)-2-methylpropanoic acid (**19d**)

The title compound was obtained similarly to **19a.** N-(3,5-dimethylphenyl)-2-(4-hydroxyphenyl)acetamide **12a** was replaced with N-(3,5-dimethylphenyl)-4-(4-hydroxyphenyl)butanamide **12d.** A yellow solid was obtained; yield: 68.9%. ^1^H NMR (400 MHz, DMSO-*d6*) δ 9.70 (s, 1H), 7.17 (s, 2H), 7.02 (d, *J* = 8.1 Hz, 2H), 6.70 (d, *J* = 8.3 Hz, 2H), 6.61 (s, 1H), 2.23 (t, *J* = 7.4 Hz, 2H), 2.17 (s, 6H), 1.83–1.74 (m, 2H), 1.41 (s, 6H). ^13^C NMR (151 MHz, DMSO-*d6*) δ 171.38, 154.20, 139.65, 137.98, 134.62, 129.31, 124.93, 118.78, 117.35, 79.07, 36.28, 34.24, 27.40, 25.78, 21.53. HRMS (ESI) *m*/*z* [M + H] + calculated for C_22_H_27_NO_4_: 370.2018; found: 370.2013.

#### 4.7.5. 2-(4-(1-((3,5-dimethylphenyl)carbamoyl)cyclopropyl)phenoxy)-2-methylpropanoic acid (**19e**)

The title compound was obtained similarly to **19a.** N-(3,5-dimethylphenyl)-4-hydroxybenzamide **12b** was replaced with N-(3,5-dimethylphenyl)-1-(4-hydroxyphenyl)cyclopropane-1-carboxamide **12e.** A white solid was obtained; yield: 81.3%. ^1^H NMR (400 MHz, DMSO-*d6*) δ 8.51 (s, 1H), 7.20 (d, *J* = 8.4 Hz, 2H), 7.07 (s, 2H), 6.75 (d, *J* = 8.5 Hz, 2H), 6.62 (s, 1H), 2.14 (s, 6H), 1.41 (s, 6H), 1.31 (dd, *J* = 6.5, 4.0 Hz, 2H), 0.98 (dd, *J* = 6.6, 4.1 Hz, 2H). ^13^C NMR (151 MHz, DMSO-*d6*) δ 175.54, 171.60, 154.78, 139.01, 137.91, 133.14, 130.67, 125.46, 118.79, 118.27, 79.04, 31.26, 25.66, 21.50, 15.19. HRMS (ESI) *m*/*z* [M + H] + calculated for C_22_H_25_NO_4_: 368.1862; found: 368.1856.

#### 4.7.6. 2-methyl-2-(4-(2-oxo-2-(o-tolylamino)ethyl)phenoxy)propanoic acid(**19f**)

The title compound was obtained similarly to **19a.** N-(3,5-dimethylphenyl)-2-(4-hydroxyphenyl)acetamide **12a** was replaced with 2-(4-hydroxyphenyl)-N-(o-tolyl)acetamide **12f.** A pale yellow solid was obtained; yield: 66.3%. ^1^H NMR (600 MHz, DMSO-*d6*) δ 13.00 (s, 1H), 9.41 (s, 1H), 7.38 (t, *J* = 6.8 Hz, 1H), 7.24 (d, *J* = 8.5 Hz, 2H), 7.19 (d, *J* = 7.5 Hz, 1H), 7.15–7.11 (m, 1H), 7.06 (t, *J* = 7.2 Hz, 1H), 6.79 (d, *J* = 8.5 Hz, 2H), 3.58 (s, 2H), 2.14 (d, *J* = 5.1 Hz, 3H), 1.49 (s, 6H). ^13^C NMR (151 MHz, DMSO-*d6*) δ 176.24, 170.15, 156.54, 136.79, 132.18, 130.71, 130.42, 129.51, 126.78, 126.36, 118.11, 115.59, 80.34, 42.43, 26.50, 18.21. HRMS (ESI) *m*/*z* [M + H] + calculated for C_19_H_21_NO_4_: 328.1549; found:328.1543.

#### 4.7.7. N-(4-bromophenyl)-2-(4-((1-hydroxy-2-methylpropan-2-yl)oxy)phenyl)acetamide (**19g**)

The title compound was obtained similarly to **19a.** N-(3,5-dimethylphenyl)-2-(4-hydroxyphenyl)acetamide **12a** was replaced with N-(4-bromophenyl)-2-(4-hydroxyphenyl)acetamide **12g.** A white solid was obtained; yield: 85.3%. ^1^H NMR (400 MHz, DMSO-*d6*) δ 10.68 (s, 1H), 7.62–7.56 (m, 2H), 7.44–7.39 (m, 2H), 7.04 (d, *J* = 8.5 Hz, 2H), 6.71 (d, *J* = 8.6 Hz, 2H), 3.48 (s, 2H), 1.34 (s, 6H). ^13^C NMR (151 MHz, DMSO-*d6*) δ 176.28, 170.09, 154.90, 139.13, 131.95, 128.67, 121.47, 118.66, 115.12, 79.25, 42.87, 25.81. HRMS (ESI) *m*/*z* [M + H] + calculated for C_18_H_18_BrNO_4_: 392.0497; found:392.0492.

#### 4.7.8. 2-(4-(2-((3,5-dibromophenyl)amino)-2-oxoethyl)phenoxy)-2-methylpropanoic acid (**19h**)

The title compound was obtained similarly to **19a.** N-(3,5-dimethylphenyl)-2-(4-hydroxyphenyl)acetamide **12a** was replaced with N-(3,5-dibromophenyl)-2-(4-hydroxyphenyl)acetamide **12h.** A brown solid was obtained; yield: 87.7%. ^1^H NMR (400 MHz, DMSO-*d6*) δ 11.56 (s, 1H), 7.94 (d, *J* = 1.7 Hz, 2H), 7.40–7.38 (m, 1H), 7.05 (d, *J* = 8.6 Hz, 2H), 6.72 (d, *J* = 8.6 Hz, 2H), 3.53 (s, 2H), 1.37 (s, 6H). ^13^C NMR (151 MHz, DMSO-*d6*) δ 176.77, 171.22, 156.00, 142.90, 129.76, 127.72, 126.99, 122.65, 120.93, 117.94, 80.40, 42.88, 26.62. HRMS (ESI) *m*/*z* [M + H] + calculated for C_18_H_17_Br_2_NO_4_: 471.9582; found: 471.9584.

#### 4.7.9. 2-(4-(2-((3-iodophenyl)amino)-2-oxoethyl)phenoxy)-2-methylpropanoic acid (**19i**)

The title compound was obtained similarly to **19a.** N-(3,5-dimethylphenyl)-4-hydroxybenzamide **12b** was replaced with2-(4-hydroxyphenyl)-N-(4-iodophenyl)acetamide **12i.** A yellow solid was obtained; yield: 82.1%. ^1^H NMR (400 MHz, DMSO-*d6*) δ 10.72 (s, 1H), 8.10 (t, *J* = 1.8 Hz, 1H), 7.55 (ddd, *J* = 8.2, 2.0, 0.9 Hz, 1H), 7.31 (ddd, *J* = 7.8, 1.7, 1.0 Hz, 1H), 7.05–6.99 (m, 3H), 6.71 (d, *J* = 8.6 Hz, 2H), 3.47 (s, 2H), 1.35 (d, *J* = 8.2 Hz, 6H). ^13^C NMR (151 MHz, DMSO-*d6*) δ 176.82, 170.63, 156.00, 141.36, 131.99, 131.16, 129.69, 127.74, 127.06, 118.82, 118.03, 94.92, 80.49, 42.91, 26.61. HRMS (ESI) *m*/*z* [M + H] + calculated for C_18_H_18_INO_4_: 440.0359; found: 440.0353.

#### 4.7.10. 2-(4-(2-((3-. bromo-5-chlorophenyl)amino)-2-oxoethyl)phenoxy)-2-methylpropanoic acid (**19j**)

The title compound was obtained similarly to **19a.** N-(3,5-dimethylphenyl)-2-(4-hydroxyphenyl)acetamide **12a** was replaced with N-(3-bromo-5-chlorophenyl)-2-(4-hydroxyphenyl)acetamide **12j.** A brown solid was obtained; yield: 61.0%. ^1^H NMR (400 MHz, DMSO-*d6*) δ 13.05–12.76 (m, 1H), 10.42 (s, 1H), 7.76 (t, *J* = 1.8 Hz, 1H), 7.66 (dd, *J* = 2.3, 1.4 Hz, 1H), 7.34 (dd, *J* = 3.9, 2.1 Hz, 1H), 7.18–7.13 (m, 2H), 6.76–6.71 (m, 2H), 3.53 (s, 2H), 1.45 (s, 6H). ^13^C NMR (151 MHz, DMSO-*d6*) δ 175.53, 170.61, 154.59, 142.04, 134.70, 130.41, 128.79, 125.60, 122.62, 118.80, 118.01, 78.79, 42.82. HRMS (ESI) *m*/*z* [M + H] + calculated for C_18_H_17_BrClNO_4_: 426.0107; found: 426.0102.

#### 4.7.11. 2-(4-(2-((2-bromophenyl)amino)-2-oxoethyl)phenoxy)-2-methylpropanoic acid (**19k**)

The title compound was obtained similarly to **19a.** N-(3,5-dimethylphenyl)-2-(4-hydroxyphenyl)acetamide **12a** was replaced with N-(2-bromophenyl)-2-(4-hydroxyphenyl)acetamide **12k.** A yellow solid was obtained; yield: 31.8%. ^1^H NMR (600 MHz, DMSO-*d6*) δ 12.97 (s, 1H), 9.49 (s, 1H), 7.64 (dq, *J* = 7.4, 1.7 Hz, 2H), 7.39–7.29 (m, 1H), 7.26 (d, *J* = 8.3 Hz, 2H), 7.14–7.08 (m, 1H), 6.79 (t, *J* = 10.9 Hz, 2H), 3.67 (d, *J* = 30.1 Hz, 2H), 1.49 (d, *J* = 11.9 Hz, 6H). ^13^C NMR (151 MHz, DMSO-*d6*) δ 175.51, 170.00, 154.54, 136.62, 133.08, 130.52, 129.30, 128.46, 127.37, 127.17, 118.92, 117.94, 78.82, 42.20, 25.53. HRMS (ESI) *m*/*z* [M + H] + calculated for C_18_H_18_BrNO_4_: 392.0497; found: 392.0492.

#### 4.7.12. 2-(4-(2-((2-bromo-3-methylphenyl)amino)-2-oxoethyl)phenoxy)-2-methylpropanoic acid (**18l**)

The title compound was obtained similarly to **19a.** N-(3,5-dimethylphenyl)-4-hydroxybenzamide **12b** was replaced with N-(2-bromo-3-methylphenyl)-2-(4-hydroxyphenyl)acetamide **12l.** A pale yellow solid was obtained; yield: 89.5%. ^1^H NMR (400 MHz, DMSO-*d6*) δ 9.78 (s, 1H), 7.39 (d, *J* = 8.0 Hz, 1H), 7.28 (d, *J* = 8.0 Hz, 1H), 7.10 (dd, *J* = 18.4, 5.7 Hz, 2H), 7.03 (dd, *J* = 12.8, 8.1 Hz, 1H), 6.71 (dd, *J* = 14.9, 8.5 Hz, 2H), 3.52 (s, 2H), 2.21–2.06 (m, 3H), 1.37 (d, *J* = 12.6 Hz, 6H). ^13^C NMR (151 MHz, DMSO-*d6*) δ 175.95, 170.11, 155.08, 138.24, 132.67, 130.03, 129.78, 128.66, 127.70, 125.79, 125.12, 118.60, 79.43, 42.18, 25.96, 18.49. HRMS (ESI) *m*/*z* [M + H] + calculated for C_19_H_20_BrNO_4_: 406.0654; found: 406.0648.

#### 4.7.13. 2-methyl-2-(4-(2-((4-(methylthio)phenyl)amino)-2-oxoethyl)phenoxy)propanoic acid (**19m**)

The title compound was obtained similarly to **19a.** N-(3,5-dimethylphenyl)-4-hydroxybenzamide **12b** was replaced with 2-(4-hydroxyphenyl)-N-(4-(methylthio)phenyl)acetamide **12m.** A brown solid was obtained; yield: 84.6%. ^1^H NMR (600 MHz, DMSO-*d6*) δ 12.99 (s, 1H), 10.11 (s, 1H), 7.62–7.48 (m, 2H), 7.21 (dt, *J* = 12.2, 4.1 Hz, 4H), 6.81–6.73 (m, 2H), 3.54 (s, 2H), 2.43 (s, 3H), 1.49 (s, 6H). ^13^C NMR (151 MHz, DMSO-*d6*) δ 175.51, 169.70, 154.48, 137.20, 132.16, 130.33, 129.48, 127.58, 120.26, 118.78, 78.78, 43.24, 25.52, 16.00. HRMS (ESI) *m*/*z* [M + H] + calculated for C_19_H_21_NO_4_S: 360.1269; found: 360.1264.

#### 4.7.14. 2-methyl-2-(4-(2-((3-(methylthio)phenyl)amino)-2-oxoethyl)phenoxy)propanoic acid (**19m-1**)

The title compound was obtained similarly to **19a.** N-(3,5-dimethylphenyl)-4-hydroxybenzamide **12b** was replaced with 2-(4-hydroxyphenyl)-N-(3-(methylthio)phenyl)acetamide **12m-1.** A white solid was obtained; yield: 38.3%. ^1^H NMR (400 MHz, DMSO-*d6*) δ 10.57 (s, 1H), 7.59 (t, *J* = 1.9 Hz, 1H), 7.37–7.31 (m, 1H), 7.15 (dd, *J* = 10.3, 5.6 Hz, 1H), 7.04 (d, *J* = 8.7 Hz, 2H), 6.85 (ddd, *J* = 7.9, 1.9, 0.9 Hz, 1H), 6.74–6.68 (m, 2H), 3.47 (s, 2H), 2.39 (s, 3H), 1.34 (s, 6H). ^13^C NMR (151 MHz, DMSO-*d6*) δ 175.73, 170.09, 154.93, 140.30, 138.99, 130.16, 129.66, 128.73, 120.96, 118.58, 116.53, 116.02, 79.23, 42.92, 25.82, 15.04. HRMS (ESI) *m*/*z* [M + H] + calculated for C_19_H_21_NO_4_S: 360.1269; found: 360.1264.

#### 4.7.15. 2-(4-(2-((4-ethynylphenyl)amino)-2-oxoethyl)phenoxy)-2-methylpropanoic acid (**19n**)

The title compound was obtained similarly to **19a.** N-(3,5-dimethylphenyl)-4-hydroxybenzamide **12b** was replaced with N-(4-ethynylphenyl)-2-(4-hydroxyphenyl)acetamide **12n.** A brown solid was obtained; yield: 41.5%. ^1^H NMR (600 MHz, DMSO-*d6*) δ 12.94 (s, 1H), 10.28 (s, 1H), 7.61 (d, *J* = 8.7 Hz, 2H), 7.40 (d, *J* = 8.7 Hz, 2H), 7.20 (d, *J* = 8.6 Hz, 2H), 6.78 (d, *J* = 8.6 Hz, 2H), 4.06 (s, 1H), 3.57 (s, 2H), 1.49 (s, 6H). ^13^C NMR (151 MHz, DMSO-*d6*) δ 175.49, 170.05, 154.53, 140.25, 132.83, 130.37, 129.26, 119.37, 118.79, 116.53, 84.02, 80.28, 78.78, 42.87, 25.53. HRMS (ESI) *m*/*z* [M + H] + calculated for C_20_H_19_NO_4_: 338.1392; found: 338.1387.

#### 4.7.16. 2-(4-(2-((3-ethynylphenyl)amino)-2-oxoethyl)phenoxy)-2-methylpropanoic acid (**19n-1**)

The title compound was obtained similarly to **19a.** N-(3,5-dimethylphenyl)-4-hydroxybenzamide **12b** was replaced with N-(3-ethynylphenyl)-2-(4-hydroxyphenyl)acetamide **12n-1.** A brown solid was obtained; yield: 54.9%. ^1^H NMR (600 MHz, DMSO-*d6*)δ 12.93 (s, 1H), 10.22 (s, 1H), 7.78 (t, *J* = 1.6 Hz, 1H), 7.56 (dd, *J* = 8.3, 1.1 Hz, 1H), 7.31 (t, *J* = 7.9 Hz, 1H), 7.21 (d, *J* = 8.6 Hz, 2H), 7.14 (dd, *J* = 6.5, 1.2 Hz, 1H), 6.80–6.76 (m, 2H), 4.15 (s, 1H), 3.56 (s, 2H), 1.49 (s, 6H). ^13^C NMR (151 MHz, DMSO-*d6*) δ 173.21, 167.79, 152.24, 137.60, 128.07, 127.38, 126.99, 124.60, 120.18, 120.08, 117.84, 116.51, 81.54, 78.71, 76.49, 40.56, 23.24. HRMS (ESI) *m*/*z* [M + H] + calculated for C_20_H_19_NO_4_: 338.1392; found: 338.1387.

#### 4.7.17. 2-(4-(2-([1,1′-biphenyl]-4-ylamino)-2-oxoethyl)phenoxy)-2-methylpropanoic acid (**19o**)

The title compound was obtained similarly to **19a.** N-(3,5-dimethylphenyl)-4-hydroxybenzamide **12b** was replaced with N-([1,1′-biphenyl]-4-yl)-2-(4-hydroxyphenyl)acetamide **12o.** A white solid was obtained; yield: 97.8%. ^1^H NMR (600 MHz, DMSO-*d6*) δ 12.95 (s, 1H), 10.21 (s, 1H), 7.69 (d, *J* = 8.7 Hz, 2H), 7.62 (ddd, *J* = 8.7, 7.6, 1.5 Hz, 4H), 7.43 (t, *J* = 7.8 Hz, 2H), 7.32 (t, *J* = 7.4 Hz, 1H), 7.23 (d, *J* = 8.6 Hz, 2H), 6.81–6.77 (m, 2H), 3.58 (s, 2H), 1.49 (s, 6H). ^13^C NMR (151 MHz, DMSO-*d6*) δ 175.51, 169.84, 154.51, 140.17, 139.20, 135.27, 130.36, 129.50, 129.36, 127.47, 127.37, 126.69, 119.93, 118.82, 78.79, 42.98, 25.54. HRMS (ESI) *m*/*z* [M + H] + calculated for C_24_H_23_NO_4_: 390.1705; found: 390.1700.

#### 4.7.18. 2-(4-(2-([1,1′-biphenyl]-3-ylamino)-2-oxoethyl)phenoxy)-2-methylpropanoic acid (**19p**)

The title compound was obtained similarly to **19a.** N-(3,5-dimethylphenyl)-4-hydroxybenzamide **12b** was replaced with N-([1,1′-biphenyl]-3-yl)-2-(4-hydroxyphenyl)acetamide **12p.** A white solid was obtained; yield: 51.4%. ^1^H NMR (600 MHz, DMSO-*d6*) δ 12.91 (s, 1H), 10.22 (s, 1H), 7.93 (t, *J* = 1.7 Hz, 1H), 7.62–7.55 (m, 1H), 7.47 (dd, *J* = 10.6, 4.9 Hz, 1H), 7.38 (dt, *J* = 8.9, 7.7 Hz, 1H), 7.32 (d, *J* = 7.8 Hz, 1H), 7.23 (d, *J* = 8.6 Hz, 1H), 6.81–6.77 (m, 1H), 3.58 (s, 1H), 1.49 (s, 6H). ^13^C NMR (151 MHz, DMSO-*d6*) δ 175.51, 169.96, 154.51, 141.23, 140.62, 140.30, 130.39, 129.81, 129.46, 129.43, 128.03, 127.09, 122.06, 118.82, 118.57, 117.85, 78.79, 42.94, 25.54. HRMS (ESI) *m*/*z* [M + H] + calculated for C_24_H_23_NO_4_: 390.1705; found: 390.1700.

#### 4.7.19. 2-(4-(2-((4-cyanophenyl)amino)-2-oxoethyl)phenoxy)-2-methylpropanoic acid (**19q)**

The title compound was obtained similarly to **19a.** N-(3,5-dimethylphenyl)-4-hydroxybenzamide **12b** was replaced with N-(4-cyanophenyl)-2-(4-hydroxyphenyl)acetamide **12q.** A white solid was obtained; yield: 64.9%. ^1^H NMR (400 MHz, DMSO-*d6*) δ 12.98 (s, 1H), 10.55 (s, 1H), 7.77–7.65 (m, 4H), 7.19–7.13 (m, 2H), 6.76–6.70 (m, 2H), 3.57 (s, 2H), 1.45 (s, 6H). ^13^C NMR (151 MHz, DMSO-*d6*) δ 175.50, 170.77, 154.53, 143.84, 133.83, 131.15, 128.88, 119.59, 119.53, 118.79, 105.39, 78.78, 42.94, 25.52. HRMS (ESI) *m*/*z* [M + H] + calculated for C_19_H_18_N_2_O_4_: 339.1345; found: 339.1340.

#### 4.7.20. 2-(4-(2-((4-(diethylamino)phenyl)amino)-2-oxoethyl)phenoxy)-2-methylpropanoic acid (**19r**)

The title compound was obtained similarly to **19a.** N-(3,5-dimethylphenyl)-2-(4-hydroxyphenyl)acetamide **12a** was replaced with N-(4-(diethylamino)phenyl)-2-(4-hydroxyphenyl)acetamide **12r.** A white brown was obtained; yield: 42.6%. ^1^H NMR (400 MHz, DMSO-*d6*) δ 9.77 (s, 1H), 7.31 (d, *J* = 9.0 Hz, 2H), 7.04 (d, *J* = 8.1 Hz, 2H), 6.71 (d, *J* = 8.1 Hz, 2H), 6.54 (d, *J* = 9.0 Hz, 2H), 3.38 (s, 2H), 3.22 (dd, *J* = 14.0, 7.1 Hz, 4H), 1.34 (s, 6H), 0.99 (t, *J* = 7.0 Hz, 6H). ^13^C NMR (151 MHz, DMSO-*d6*) δ 176.44, 169.10, 155.38, 144.42, 129.77, 128.58, 121.47, 118.30, 112.45, 112.29, 44.24, 42.81, 26.23, 12.82. HRMS (ESI) *m*/*z* [M + H] + calculated for C_22_H_28_N_2_O_4_: 385.2127; found: 385.2122.

#### 4.7.21. 2-(4-(2-((4-methoxyphenyl)amino)-2-oxoethyl)phenoxy)-2-methylpropanoic acid (**19s**)

The title compound was obtained similarly to **19a.** N-(3,5-dimethylphenyl)-4-hydroxybenzamide **12b** was replaced with 2-(4-hydroxyphenyl)-N-(4-methoxyphenyl)acetamide **12s.** A white solid was obtained; yield: 47.1%. ^1^H NMR (400 MHz, DMSO-*d6*) δ 12.99 (s, 1H), 9.95 (s, 1H), 7.49–7.43 (m, 2H), 7.16 (d, *J* = 8.6 Hz, 2H), 6.85–6.80 (m, 2H), 6.77–6.71 (m, 2H), 3.67 (s, 3H), 3.47 (s, 2H), 1.45 (s, 6H). ^13^C NMR (151 MHz, DMSO-*d6*) δ 175.51, 169.23, 155.58, 154.45, 132.92, 130.29, 129.70, 121.06, 118.79, 114.28, 78.78, 55.60, 42.83, 25.53. HRMS (ESI) *m*/*z* [M + H] + calculated for C_19_H_21_NO_5_: 344.1498; found: 344.1492.

#### 4.7.22. (E)-2-methyl-2-(4-(3-((3-(methylthio)phenyl)amino)-3-oxoprop-1-en-1- yl)phenoxy) propanoic acid (**19t**)

The title compound was obtained similarly to **19b.** N-(3,5-dimethylphenyl)-4-hydroxybenzamide **12b** was replaced with (E)-3-(4-hydroxyphenyl)-N-(3-(methylthio)phenyl)acrylamide **12t.** A white solid was obtained; yield: 91.6%. ^1^H NMR (600 MHz, DMSO-*d6*) δ 13.14 (s, 1H), 10.15 (s, 1H), 7.69 (t, *J* = 1.8 Hz, 1H), 7.53 (t, *J* = 11.5 Hz, 3H), 7.42 (dd, *J* = 8.1, 1.1 Hz, 1H), 7.26 (t, *J* = 8.0 Hz, 1H), 6.96–6.94 (m, 1H), 6.86 (dd, *J* = 9.2, 2.4 Hz, 2H), 6.66 (d, *J* = 15.7 Hz, 1H), 2.47 (s, 3H), 1.55 (s, 6H). ^13^C NMR (151 MHz, DMSO-*d6*) δ 175.24, 164.40, 157.32, 129.71, 129.58, 128.30, 121.06, 120.42, 118.64, 116.64, 116.16, 79.09, 25.56, 15.06. HRMS (ESI) *m*/*z* [M + H] + calculated for C_20_H_21_NO_4_S: 372.1269; found: 372.1264.

#### 4.7.23. (E)-2-(4-(3-((4-bromophenyl)amino)-3-oxoprop-1-en-1-yl)phenoxy)-2-methylpropanoic acid (**19u**)

The title compound was obtained similarly to **19b.** N-(3,5-dimethylphenyl)-4-hydroxybenzamide **12b** was replaced with (E)-N-(4-bromophenyl)-3-(4-hydroxyphenyl)acrylamide **12u.** A white solid was obtained; yield: 70.8%. ^1^H NMR (600 MHz, DMSO-*d6*)δ 13.08 (s, 1H), 10.27 (s, 1H), 7.69–7.64 (m, 2H), 7.57–7.49 (m, 5H), 6.86 (dd, *J* = 6.8, 4.9 Hz, 2H), 6.65 (d, *J* = 15.7 Hz, 1H), 1.55 (s, 6H). ^13^C NMR (151 MHz, DMSO-*d6*) δ 175.23, 164.40, 157.36, 140.63, 139.18, 132.06, 129.61, 128.25, 121.57, 120.25, 118.63, 115.28, 79.09, 25.56. HRMS (ESI) *m*/*z* [M + H] + calculated for C_19_H_18_BrNO_4_: 404.0497; found: 404.0492.

#### 4.7.24. (E)-2-(4-(3-((3-iodophenyl)amino)-3-oxoprop-1-en-1-yl)phenoxy)-2-methylpropanoic acid (**19v**)

The title compound was obtained similarly to **19b.** N-(3,5-dimethylphenyl)-4-hydroxybenzamide **12b** was replaced with (E)-3-(4-hydroxyphenyl)-N-(3-iodophenyl)acrylamide **12v.** A white solid was obtained; yield: 65.8%. ^1^H NMR (600 MHz, DMSO-*d6*)δ 13.08 (s, 1H), 10.23 (s, 1H), 8.21 (t, *J* = 1.8 Hz, 1H), 7.62 (dd, *J* = 8.2, 1.1 Hz, 1H), 7.54 (t, *J* = 11.5 Hz, 3H), 7.41 (ddd, *J* = 7.8, 1.5, 0.9 Hz, 1H), 7.13 (t, *J* = 8.0 Hz, 1H), 6.86 (dd, *J* = 9.2, 2.3 Hz, 2H), 6.64 (d, *J* = 15.7 Hz, 1H), 1.55 (s, 6H). ^13^C NMR (151 MHz, DMSO-*d6*) δ 175.22, 164.43, 157.40, 141.28, 140.74, 132.20, 131.31, 129.64, 128.20, 127.79, 120.17, 118.84, 118.63, 95.14, 79.10, 25.57. HRMS (ESI) *m*/*z* [M + H] + calculated for C_19_H_18_INO_4_: 452.0359; found: 452.0354.

#### 4.7.25. (E)-2-(4-(3-((3-bromo-5-chlorophenyl)amino)-3-oxoprop-1-en-1-yl)phenoxy)-2-methylpropanoic acid (**19w**)

The title compound was obtained similarly to **19b.** N-(3,5-dimethylphenyl)-4-hydroxybenzamide **12b** was replaced with (E)-N-(3-bromo-5-chlorophenyl)-3-(4-hydroxyphenyl)acrylamide **12w.** A brown solid was obtained; yield: 71.2%. ^1^H NMR (600 MHz, DMSO-*d6*) δ 13.14 (s, 1H), 10.46 (s, 1H), 7.89 (t, *J* = 1.7 Hz, 1H), 7.80 (t, *J* = 1.8 Hz, 1H), 7.57 (dd, *J* = 12.2, 3.4 Hz, 3H), 7.39 (t, *J* = 1.8 Hz, 1H), 6.86 (d, *J* = 8.7 Hz, 2H), 6.60 (d, *J* = 15.7 Hz, 1H), 1.56 (s, 6H). ^13^C NMR (151 MHz, DMSO-*d6*) δ 175.21, 164.80, 157.59, 142.31, 141.50, 134.72, 129.80, 127.94, 125.51, 122.65 (s), 120.56, 119.60, 118.59, 118.08, 79.03, 25.81–25.06. HRMS (ESI) *m*/*z* [M + H] + calculated for C_19_H_17_BrClNO_4_: 438.0107; found: 438.0102.

#### 4.7.26. (E)-2-(4-(3-((3,5-dimethylphenyl)amino)-3-oxoprop-1-en-1-yl)-2-methoxyphenoxy)-2-methylpropanoic acid (**19x**)

The title compound was obtained similarly to **19b.** N-(3,5-dimethylphenyl)-4-hydroxybenzamide **12b** was replaced with (E)-N-(3,5-dimethylphenyl)-3-(4-hydroxy-3-methoxyphenyl)acrylamide **12x.** A white solid was obtained; yield: 45.9%. ^1^H NMR (400 MHz, CHLOROFORM-D) δ 7.69 (dd, *J* = 15.8, 9.7 Hz, 2H), 7.14–7.06 (m, 3H), 7.01 (dd, *J* = 9.6, 5.7 Hz, 1H), 6.79 (d, *J* = 6.6 Hz, 1H), 6.46 (d, *J* = 15.2 Hz, 1H), 3.92 (d, *J* = 2.1 Hz, 3H), 2.31 (s, 6H), 1.54 (s, 6H). ^13^C NMR (151 MHz, DMSO-*d6*) δ 174.21, 163.00, 150.40, 145.40, 139.18, 138.66, 137.09, 128.40, 124.20, 120.18, 120.15, 117.79, 116.23, 110.43, 78.71, 54.85, 24.24, 20.55. HRMS (ESI) *m*/*z* [M + H] + calculated for C_22_H_25_NO_5_: 384.1811; found: 384.1806.

### 4.8. 7-(2-((3,5-dimethylphenyl)amino)-2-oxoethyl)-2,3-dihydrobenzo[b][1,4]dioxine-2-carboxylic acid (***20***)

To a stirred solution of 2-(3,4-dihydroxyphenyl)-N-(3,5-dimethylphenyl)acetamide **12y** (271 mg, 1 mmol) in 2-butanone (5 mL) was added NaOH (220 mg, 5.5 mmol). The reaction mixture was heated to 50℃ for 1 h, and then 2,3-Dibromopropionic acid (393 mg, 1.7 mmol) was added dropwise. Afterwards, the solution was stirred for another 6 h. After completion, the reaction mixture was diluted with EA (10 mL) and acidified with 1N HCl (pH < 3). The organic phase was washed with brine (3 × 10 mL), dried over anhydrous Na_2_SO_4_ and concentrated. The residue was diluted with EA (1 mL) and PE (4 mL) under ice bath. The precipitate was collected by filtration and dried over vacuum oven. A brown solid was obtained (yield: 200 mg, 58.7%). ^1^H NMR (400 MHz, DMSO-*d6*) δ 13.27 (d, *J* = 28.5 Hz, 1H), 9.90 (d, *J* = 5.6 Hz, 1H), 7.16 (d, *J* = 2.9 Hz, 2H), 6.86–6.81 (m, 1H), 6.78–6.74 (m, 1H), 6.74–6.70 (m, 1H), 6.63 (d, *J* = 0.6 Hz, 1H), 4.98 (dt, *J* = 7.0, 3.1 Hz, 1H), 4.37 (ddd, *J* = 11.5, 3.4, 1.3 Hz, 1H), 4.19 (ddd, *J* = 11.6, 7.6, 2.8 Hz, 1H), 3.43 (d, *J* = 7.3 Hz, 2H), 2.17 (d, *J* = 0.6 Hz, 6H). ^13^C NMR (151 MHz, DMSO-*d6*) δ 168.98, 168.45, 141.53, 141.10, 138.47, 137.03, 129.02, 124.10, 121.76, 121.11, 116.78, 116.27, 70.73, 64.19, 42.04, 20.48. HRMS (ESI) *m*/*z* [M + H] + calculated for C_19_H_19_NO_5_: 342.1341; found: 342.1336.

### 4.9. Procedure for the Syntheses of Intermediates ***23a–23g***

#### 4.9.1. N,N-diallyl-2-(4-hydroxyphenyl)acetamide (**23a**)

To a stirring solution of 4-hydroxybenzoic acid **11a** (0.73 g, 6 mmol), EDCI (1.44 g, 7.5 mmol) and HOBT (0.81 g, 6 mmol) in dry DCM (20 mL) at room temperature was added diallylamine **22a** (0.58 g, 6 mmol) for 8 h. Upon completion, the mixture was washed with water (3 × 20 mL) and dried over anhydrous sodium sulfate before concentration in vacuo. The crude product was purified with silica gel column chromatography using PE/EA (4:1 to 2:1) as eluent to afford **23a** as a white solid (0.83 g, yield: 71.9%). ^1^H NMR (600 MHz, DMSO-*d6*) δ 9.22 (s, 1H), 7.00 (d, *J* = 8.5 Hz, 2H), 6.70–6.66 (m, 2H), 5.72 (qdd, *J* = 21.6, 10.7, 5.4 Hz, 2H), 5.18–5.03 (m, 4H), 3.87 (dd, *J* = 21.7, 5.4 Hz, 4H), 3.53 (s, 2H). ESI-MS: *m*/*z* = 232.13 [M + H]^+^.

#### 4.9.2. 2-(4-hydroxyphenyl)-1-(2-methylindolin-1-yl)ethan-1-one (**23b**)

The title compound was obtained similarly to **23a.** Diallylamine **22a** was replaced with 2-methylindoline **22b.** A white solid was obtained; yield: 30.1%. ^1^H NMR (400 MHz, DMSO-*d6*) δ 9.23 (s, 1H), 7.93 (t, *J* = 16.9 Hz, 1H), 7.21 (d, *J* = 7.1 Hz, 1H), 7.08 (dt, *J* = 29.6, 10.0 Hz, 3H), 6.97 (td, *J* = 7.4, 1.1 Hz, 1H), 6.70–6.58 (m, 2H), 4.74–4.62 (m, 1H), 3.79 (d, *J* = 15.2 Hz, 1H), 3.68–3.56 (m, 1H), 3.30–3.17 (m, 1H), 2.62 (t, *J* = 10.4 Hz, 1H), 1.17 (d, *J* = 6.1 Hz, 3H). ESI-MS: *m*/*z* = 268.13 [M + H]^+^.

#### 4.9.3. N-(benzo[d][1,3]dioxol-5-yl)-2-(4-hydroxyphenyl)acetamide (**23c**)

The title compound was obtained similarly to **23a.** Diallylamine **22a** was replaced with benzo[d][1,3]dioxol-5-amine **22C.** A white solid was obtained; yield: 44.1%. ^1^H NMR (400 MHz, DMSO-*d6*) δ 9.94 (s, 1H), 9.22 (s, 1H), 7.26 (d, *J* = 2.0 Hz, 1H), 7.06 (d, *J* = 8.5 Hz, 2H), 6.91 (dd, *J* = 8.4, 2.1 Hz, 1H), 6.79 (d, *J* = 8.4 Hz, 1H), 6.68–6.63 (m, 2H), 5.92 (s, 2H), 3.41 (s, 2H). ESI-MS: *m*/*z* = 272.08 [M + H]^+^.

#### 4.9.4. 2-(4-hydroxyphenyl)-N-(naphthalen-2-yl)acetamide (**23d**)

The title compound was obtained similarly to **23a.** Diallylamine **22a** was replaced with naphthalen-2-amine **22d.** A white solid was obtained; yield: 72.2%. ^1^H NMR (600 MHz, DMSO-*d6*) δ 10.26 (s, 1H), 9.25 (s, 1H), 8.28 (d, *J* = 1.6 Hz, 1H), 7.87–7.76 (m, 3H), 7.59 (dd, *J* = 8.8, 2.1 Hz, 1H), 7.42 (dtd, *J* = 16.1, 6.8, 1.1 Hz, 2H), 7.16 (d, *J* = 8.5 Hz, 2H), 6.73–6.70 (m, 2H). ESI-MS: *m*/*z* = 278.12 [M + H]^+^.

#### 4.9.5. N-benzhydryl-2-(4-hydroxyphenyl)acetamide (**23e**)

The title compound was obtained similarly to **23a.** Diallylamine **22a** was replaced with diphenylmethanamine **22e.** A white solid was obtained; yield: 30.9%. ^1^H NMR (600 MHz, DMSO-*d6*) δ 9.19 (s, 1H), 8.89 (d, *J* = 8.6 Hz, 1H), 7.32 (dd, *J* = 10.5, 4.5 Hz, 5H), 7.25 (d, *J* = 7.9 Hz, 5H), 7.05 (d, *J* = 8.4 Hz, 2H), 6.67–6.63 (m, 2H), 6.08 (d, *J* = 8.6 Hz, 1H), 3.41 (s, 2H). ESI-MS: *m*/*z* = 318.14 [M + H]^+^.

#### 4.9.6. N,N-dibenzyl-2-(4-hydroxyphenyl)acetamide (**23f**)

The title compound was obtained similarly to **23a.** Diallylamine **22a** was replaced with dibenzylamine **22f.** A white solid was obtained; yield: 30.2% ^1^H NMR (600 MHz, DMSO-*d6*) δ 9.24 (s, 1H), 7.36 (t, *J* = 7.5 Hz, 2H), 7.30 (dd, *J* = 10.0, 4.4 Hz, 3H), 7.24 (t, *J* = 7.3 Hz, 1H), 7.16 (t, *J* = 6.8 Hz, 4H), 7.00 (d, *J* = 8.5 Hz, 2H), 6.69–6.66 (m, 2H), 4.50 (d, *J* = 34.5 Hz, 4H), 3.64 (s, 2H). ESI-MS: *m*/*z* = 332.15 [M + H]^+^.

#### 4.9.7. N-hexyl-2-(4-hydroxyphenyl)acetamide (**23g**)

The title compound was obtained similarly to **23a.** Diallylamine **22a** was replaced with hexan-1-amine **22g.** A white solid was obtained; yield: 72.0% ^1^H NMR (400 MHz, DMSO-*d6*) δ 9.17 (s, 1H), 7.84 (t, *J* = 5.7 Hz, 1H), 7.01–6.95 (m, 2H), 6.65–6.58 (m, 2H), 3.19 (s, 2H), 2.95 (dd, *J* = 12.6, 6.9 Hz, 2H), 1.31 (dd, *J* = 13.7, 6.9 Hz, 2H), 1.24–1.15 (m, 6H), 0.85–0.77 (m, 3H). ESI-MS: *m*/*z* = 236.16 [M + H]^+^.

### 4.10. Procedure for the Syntheses of ***24a–24g***

#### 4.10.1. 2-(4-(2-(diallylamino)-2-oxoethyl)phenoxy)-2-methylpropanoic acid (**24a**)

To a stirred solution of N,N-diallyl-2-(4-hydroxyphenyl)acetamide **23a** (0.46 g, 2 mmol) in acetone (6 mL) was added NaOH (0.96 g, 24 mmol). The reaction mixture was cooled down to 0 ℃ for 0.5 h, and then chloroform (0.72 mg, 6 mmol) was added dropwise. Afterwards, the solution was stirred for another 6 h. After completion, the reaction mixture was diluted with EA (15 mL) and acidified with 1N HCl (pH < 3). The organic phase was washed with brine (3 × 15 mL), dried over anhydrous Na_2_SO_4_ and concentrated. The residue was diluted with EA (2 mL) and PE (8 mL) under ice bath. The precipitate was collected by filtration and dried over vacuum oven. A white solid was obtained (0.52 g, yield: 51.5%). ^1^H NMR (600 MHz, DMSO-*d6*) δ 12.97 (s, 1H), 7.09 (d, *J* = 8.6 Hz, 1H), 6.76 (d, *J* = 8.6 Hz, 1H), 5.74 (dtt, *J* = 16.0, 11.2, 5.4 Hz, 1H), 5.18–5.04 (m, 1H), 3.92 (d, *J* = 5.1 Hz, 1H), 3.86 (d, *J* = 5.7 Hz, 1H), 3.59 (s, 1H), 1.48 (s, 6H). ^13^C NMR (151 MHz, DMSO-*d6*) δ 175.56, 170.87, 154.30, 134.34, 130.35, 129.39, 118.83, 116.89, 78.80, 49.79, 47.77, 25.50. HRMS (ESI) *m*/*z* [M + H] + calculated for C_18_H_23_NO_4_: 318.1705; found: 318.1701.

#### 4.10.2. 2-methyl-2-(4-(2-(2-methylindolin-1-yl)-2-oxoethyl)phenoxy)propanoic acid (**24b**)

The title compound was obtained similarly to **24a.** 2-(4-(2-(diallylamino)-2-oxoethyl)phenoxy)-2-methylpropanoic acid **23a** was replaced with 2-(4-hydroxyphenyl)-1-(2-methylindolin-1-yl)ethan-1-one **23b.** A white solid was obtained; yield: 56.0%. ^1^H NMR (400 MHz, DMSO-*d6*) δ 12.98 (s, 1H), 7.95 (d, *J* = 7.4 Hz, 1H), 7.22 (d, *J* = 7.2 Hz, 1H), 7.13 (dd, *J* = 21.2, 8.0 Hz, 3H), 6.97 (td, *J* = 7.5, 1.0 Hz, 1H), 6.75–6.70 (m, 2H), 4.77–4.66 (m, 1H), 3.77 (dd, *J* = 74.1, 15.6 Hz, 2H), 2.62 (d, *J* = 15.4 Hz, 1H), 1.45 (s, 6H), 1.24–1.14 (m, 3H). ^13^C NMR (151 MHz, DMSO-*d6*) δ 175.54, 169.31, 154.41, 142.03, 131.35, 130.80, 129.12, 127.43, 125.69, 124.05, 118.70, 117.47, 78.77, 60.23, 55.70, 36.32, 25.54, 22.21. HRMS (ESI) *m*/*z* [M + H] + calculated for C_21_H_23_NO_4_: 354.1705; found: 354.1700.

#### 4.10.3. 2-(4-(2-(benzo[d][1,3]dioxol-5-ylamino)-2-oxoethyl)phenoxy)-2-methylpropanoic acid (**24c**)

The title compound was obtained similarly to **24a.** 2-(4-(2-(diallylamino)-2-oxoethyl)phenoxy)-2-methylpropanoic acid **23a** was replaced with N-(benzo[d][1,3]dioxol-5-yl)-2-(4-hydroxyphenyl)acetamide **23c.** A brown solid was obtained; yield: 71.3%. ^1^H NMR (400 MHz, DMSO-*d6*) δ 12.97 (d, *J* = 7.9 Hz, 1H), 10.05–9.93 (m, 1H), 7.26 (d, *J* = 2.0 Hz, 1H), 7.15 (dd, *J* = 6.8, 4.8 Hz, 2H), 6.91 (dd, *J* = 8.4, 2.1 Hz, 1H), 6.82–6.78 (m, 1H), 6.75–6.71 (m, 2H), 5.93 (s, 2H), 3.47 (s, 2H), 1.45 (s, 6H). ^13^C NMR (151 MHz, DMSO-*d6*) δ 175.50, 169.37, 154.47, 147.47, 143.26, 134.13, 130.31, 129.60, 101.74, 101.38, 78.77, 42.78, 25.53. HRMS (ESI) *m*/*z* [M + H] + calculated for C_19_H_19_NO_6_: 358.1290; found: 358.1286.

#### 4.10.4. 2-methyl-2-(4-(2-(naphthalen-2-ylamino)-2-oxoethyl)phenoxy)propanoic acid (**24d)**

The title compound was obtained similarly to **24a.** 2-(4-(2-(diallylamino)-2-oxoethyl)phenoxy)-2-methylpropanoic acid **23a** was replaced with 2-(4-hydroxyphenyl)-N-(naphthalen-2-yl)acetamide **23d.** A white solid was obtained; yield: 85.7%. ^1^H NMR (600 MHz, DMSO-*d6*) δ 12.99 (s, 1H), 10.34 (s, 1H), 8.31 (d, *J* = 1.7 Hz, 1H), 7.84 (dd, *J* = 17.3, 8.5 Hz, 2H), 7.79 (d, *J* = 8.0 Hz, 1H), 7.60 (dd, *J* = 8.8, 2.1 Hz, 1H), 7.46 (ddd, *J* = 8.2, 6.8, 1.2 Hz, 1H), 7.39 (ddd, *J* = 8.0, 6.9, 1.2 Hz, 1H), 7.29–7.23 (m, 2H), 6.83–6.79 (m, 2H), 3.63 (s, 2H), 1.50 (s, 6H). ^13^C NMR (151 MHz, DMSO-*d6*) δ 175.51, 170.07, 154.52, 137.27, 133.88, 130.38, 130.19, 129.50, 128.81, 127.90, 127.73, 126.87, 125.02, 120.40, 118.76 (d, *J* = 21.0 Hz), 115.63, 78.79, 42.94, 25.54. HRMS (ESI) *m*/*z* [M + H] + calculated for C_22_H_21_NO_4_: 364.1549; found: 364.1542.

#### 4.10.5. 2-(4-(2-(benzhydrylamino)-2-oxoethyl)phenoxy)-2-methylpropanoic acid (**24e**)

The title compound was obtained similarly to **24a.** 2-(4-(2-(diallylamino)-2-oxoethyl)phenoxy)-2-methylpropanoic acid **23a** was replaced with N-benzhydryl-2-(4-hydroxyphenyl)acetamide **23e.** A white solid was obtained; yield: 64.5%. ^1^H NMR (600 MHz, DMSO-*d6*) δ 12.94 (s, 1H), 8.94 (d, *J* = 8.6 Hz, 1H), 7.31 (dd, *J* = 10.3, 4.8 Hz, 1H), 7.27–7.21 (m, 1H), 7.15 (d, *J* = 8.6 Hz, 1H), 6.76–6.73 (m, 1H), 6.08 (d, *J* = 8.5 Hz, 1H), 3.47 (s, 1H), 1.48 (s, 1H). ^13^C NMR (151 MHz, DMSO-*d6*) δ 175.54, 170.07, 154.41, 142.96, 130.22, 130.03, 128.83, 127.72, 127.41, 118.81, 78.80, 56.44, 41.72, 25.52. HRMS (ESI) *m*/*z* [M + H] + calculated for C_25_H_25_NO_4_: 404.1862; found: 404.1856.

#### 4.10.6. 2-(4-(2-(dibenzylamino)-2-oxoethyl)phenoxy)-2-methylpropanoic acid **(24f**)

The title compound was obtained similarly to **24a.** 2-(4-(2-(diallylamino)-2-oxoethyl)phenoxy)-2-methylpropanoic acid **23a** was replaced with N,N-dibenzyl-2-(4-hydroxyphenyl)acetamide **23f.** A white solid was obtained; yield: 81.1%. ^1^H NMR (600 MHz, DMSO-*d6*) δ 7.35 (t, *J* = 7.5 Hz, 2H), 7.32–7.26 (m, 3H), 7.24 (t, *J* = 7.3 Hz, 1H), 7.15 (dd, *J* = 12.9, 7.3 Hz, 4H), 7.09 (d, *J* = 8.4 Hz, 2H), 6.75 (d, *J* = 8.6 Hz, 2H), 4.55 (s, 2H), 4.47 (s, 2H), 3.69 (s, 2H), 1.47 (s, 6H). ^13^C NMR (151 MHz, DMSO-*d6*) δ 172.37, 154.47, 137.48, 136.97, 130.15, 129.20, 128.96, 127.91, 127.10, 119.04, 79.24, 50.90, 48.51, 25.45. HRMS (ESI) *m*/*z* [M + H] + calculated for C_26_H_27_NO_4_: 418.2018; found: 418.2013.

#### 4.10.7. 2-(4-(2-(hexylamino)-2-oxoethyl)phenoxy)-2-methylpropanoic acid (**24g**)

The title compound was obtained similarly to **24a.** 2-(4-(2-(diallylamino)-2-oxoethyl)phenoxy)-2-methylpropanoic acid **23a** was replaced with N-hexyl-2-(4-hydroxyphenyl)acetamide **23g.** A white solid was obtained; yield: 57.7%. ^1^H NMR (400 MHz, DMSO-*d6*) δ 12.98 (d, *J* = 9.3 Hz, 1H), 7.93 (t, *J* = 5.5 Hz, 1H), 7.10–7.06 (m, 2H), 6.72–6.68 (m, 2H), 3.24 (s, 2H), 2.96 (dd, *J* = 12.6, 6.9 Hz, 2H), 1.44 (s, 6H), 1.32 (dd, *J* = 13.6, 6.6 Hz, 2H), 1.22–1.13 (m, 6H), 0.80 (t, *J* = 6.8 Hz, 3H). ^13^C NMR (151 MHz, DMSO-*d6*) δ 175.53, 170.51, 154.26, 130.22, 130.09, 118.76, 78.75, 42.00, 39.03, 31.42, 29.51, 26.51, 25.51, 22.52, 14.36. HRMS (ESI) *m*/*z* [M + H] + calculated for C_18_H_27_NO_4_: 322.2018; found: 322.2013.

#### 4.10.8. 2-methyl-2-(4-(2-(methylamino)-2-oxoethyl)phenoxy)propanoic acid (**24h**)

The title compound was obtained similarly to **24a.** 2-(4-(2-(diallylamino)-2-oxoethyl)phenoxy)-2-methylpropanoic acid **23a** was replaced with 2-(4-hydroxyphenyl)-N-methylacetamide **23h.** A colorless solid was obtained; yield: 33.7%. ^1^H NMR (400 MHz, DMSO-*d6*) δ 7.87 (d, *J* = 4.3 Hz, 1H), 7.03 (d, *J* = 8.5 Hz, 2H), 6.69 (d, *J* = 8.6 Hz, 2H), 3.23 (s, 2H), 2.51 (d, *J* = 4.6 Hz, 3H), 1.39 (s, 6H). ^13^C NMR (151 MHz, DMSO-*d6*) δ 176.04, 171.35, 154.91, 129.89, 129.14, 118.50, 79.39, 26.01, 21.23. HRMS (ESI) *m*/*z* [M + H] + calculated for C_13_H_17_NO_4_: 252.1236; found: 252.1230.

### 4.11. Biological Evaluations

#### 4.11.1. Red Blood Cell Separation

Eight-week-old female C57BL6J mice were purchased from Beijing Vital River Laboratory Animal Technology Co., Ltd. A total of 540 μL mouse blood was added into the 1.5 mL centrifuge tube, followed by adding 60 μL 3.8% sodium citrate. The mixture was then centrifuged at 1500 rpm for 10 min, the supernatant was carefully aspirated, and the cells were washed three times with PBS solution. After mixing packed red blood cells with Alsever’s solution in a ratio of 1:1, a blood gas analyzer (RADIOMETER ABL 90) was used to analyze the red blood cell density and hemoglobin concentration.

#### 4.11.2. In Vitro Red Blood Cell-Based Assay

All of the compounds were prepared as 200 mM stock solutions in DMSO. Oxygen equilibrium measurements were performed with the Softron analyzer (Softron Medical Products, Beijing, China) using mouse red blood cells. The samples were prepared by adding 3 mg hemoglobin equivalent of red blood cells into 3.92 mL loading buffer (100 mM NaCl, 50 mM bis-Tris at pH 7.2 and 25 °C), followed by 20 μL 10% Dimethicone as a defoamer and 20 μL 20% BSA. Afterwards, 40 μL stock solution was added, and the mixture was loaded to the sample cell and fully deoxygenated by blowing into nitrogen at 37 °C for 15 min. Then, the oxygen equilibrium curve was recorded by blowing into oxygen.

#### 4.11.3. Cell Culturing

Luciferase-transfected human glioma cells (U87-luc) and HEK293 cells were kindly provided by Dr. Nie and cultured in Dulbecco’s Modified Eagle Medium (CM10013, MacGene) with 10% fetal bovine serum (1966173C, Gibco) and 1% penicillin-streptomycin in a humidified atmosphere of 5% CO_2_ at 37 °C.

#### 4.11.4. Cell Titer Assay

Cells were plated in a 96-well plate at a density of 2000 cells/well and then treated with **19c** or **19t** at gradient concentrations of 10^1^, 10^2^, 10^3^, 10^4^, 10^5^, 10^6^and 10^7^ nM. Compounds **19c** and **19t** were prepared as the sodium salt by adding saturated sodium bicarbonate in the process of stock solution preparation. Cell viability was evaluated on the indicated days after treatment with a Cell Titer-Glo Luminescent Cell Viability Assay kit (G7572, Promega, Madison, WI, USA) according to the manufacturer’s instructions. All data were normalized to those on day 0 and presented as the mean ± standard deviation (SD).

#### 4.11.5. In Vivo Antitumor Activity Evaluation

Four-week-old female BALB/c nude mice were purchased from Beijing Vital River Laboratory Animal Technology Co., Ltd. Animals were monitored daily by certified veterinary staff and laboratory personnel. All mice used in this study were maintained in a pathogen-free barrier animal facility. The study was in compliance with all relevant ethical regulations regarding animal research. All animal experiments were approved by the Institutional Animal Care and Use Committee at the Beijing Institute of Basic Medical Sciences. The U87-luc cells were processed and resuspended into basal medium at a concentration of 10^7^ cells/mL. Next, 8 × 10^5^ U87-luc glioma cells were implanted into the right frontal lobes of each nude mouse. Ten days after tumor implantation, tumor-bearing mice were randomly divided into six groups according to the tumor size and subjected to the indicated treatment (*n* = 6). Administration of drugs (150mg/kg) or vehicle was performed once every 4 days four consecutive times by intraperitoneal injection. All radiation treatments were performed 25 min after the drug or vehicle was given. Tumors on the brains of mice were irradiated at a dosage of 3 Gy four consecutive times using a ^60^Co γ irradiator. The dose rate was 0.3 Gy/min. The bodies of the mice were shielded using lead blocks. The tumor progress was tracked by a small animal live bioluminescence imaging system (LB983NC100U, Germany). For experiments assessing survival, mice were monitored until the last mouse showed neurological signs.

#### 4.11.6. Molecular Docking

The co-crystal structure of deoxygenated hemoglobin protein (PDB code: 1G9V) was obtained from the RCSB Protein Data Bank. Docking results were obtained from Maestro interface (Schrödinger Suite, LLC, NY). The protein for docking was processed mainly by adding hydrogen atoms and charges, removing unrelated water molecules, immobilizing exact residues, and removing endogenous ligands. All other parameters were set to default values. Ligand preparation was performed as default. Docking results were visualized by Pymol Open-Source version 2.5.

#### 4.11.7. Pharmacokinetic Study in Rats

The pharmacokinetic experiments with **19c** in rats were performed by Beijing Pharmaron, Inc. All animals were maintained in the pathogen-free barrier animal facility, and the experiments were approved by the Animal Care and Use Committee of the company. The drug solution was formulated into a 10 mg/mL solution with normal saline on the day of administration. After a single dose of **19c** (100 mg/kg) by intraperitoneal injection (ip) or oral administration (po) in male Sprague-Dawley rats (*n* = 3), blood samples were collected 0.25, 0.5, 1, 2, 4, 6, 8 and 24 h post dose and centrifuged at 4000× *g* for 5 min at 4 °C to obtain plasma. The plasma was diluted 3 times with water. A measure of 2 µL of diluted supernatant was injected into the LC/MS/MS system for quantitative analysis. No abnormal clinical symptoms were observed during the entire experiment. The Phoenix WinNonlin 8.0 software was applied for the calculation of pharmacokinetic parameters.

#### 4.11.8. Statistical Analysis

All data were analyzed based on at least three independent experiments unless otherwise specified and shown as the mean ± SD. Data obtained from two groups at a single time point were analyzed by unpaired *t*-tests. Measurements at multiple time points were analyzed by two-way analysis of variance (ANOVA). Survival analyses were performed by the Kaplan–Meier method, with the log-rank test for comparison. The GraphPad Prism 7.0 software was used for all of the statistical analyses in this study.

## Data Availability

All data are contained within the article or Appendix A. The numerical data represented in the Figures are available on request from the corresponding author.

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
