# Peer review of "Phenoxyaromatic Acid Analogues as Novel Radiotherapy Sensitizers: Design, Synthesis and Biological Evaluation"

_molecules, 2022, doi:10.3390/molecules27082428_

Round 1
Reviewer 1 Report
This study synthesized phenoxyaromatic acid analogues that have the potential to decrease tumor hypoxia and radiosensitize GB.
Although the rationale is clear and the study includes synthesis, in vitro and in vivo validations, the manuscript would improve from extensive English writing editing and rephrasing results and conclusions. A figure on the working mechanism would be beneficial.
Is there any data on the effects of the new analogues on anti-hypoxic effects? In vivo part does not include any hypoxia validations (e.g. hypoxia IHC on brain tumor of HIF1alpha) or for example PET imaging with a hypoxia tracer. Only survival is included while the goal of the drug is to decrease hypoxia, which is not confirmed in vivo.
Specific comments:
- Throughout manuscript: Past, present and future tenses are mixed. Rephrasing needed.
- Abstract: Line 15: brain tumours is very broad maybe specific only for GB
- Abstract: Line 17: would rather use the term radioresistance. Line 18: RT sensitizers that increase the oxygen concentration within the tumor are promising to increase the effectiveness of radiation
- Abstract line 27: add tumor type: GB?
- Can abbreviate glioblastoma as GB throughout the manuscript
- Put references always at the end of the sentences
- Line 43: change to ‘the blood brain barrier (BBB) prevents’
- Line 45: In GB, the inability of chemotherapeutic drugs to pass the BBB has resulted in poor therapeutic effects (rather than the development? Or the necessity of BBB passage complicates the design of chemotherapeutic drugs). Explain why allosteric modulators are able to pass the BBB? (size, lipid,…?)
- Line 47: they have the ability to increase the oxygen supply to…
- Line 62: T state of hemoglobin
- Line 64: change ‘it’ to the drug name
- Line 66: change to ‘a’ therapeutic effect
- Line 68-69: rephrase – also not sure if applicable for all tumors or was to goal to synthesize a radiosensitizer specifically for GB? Please clarify
- Line 76: was studied in vitro
- Line 77. Rephrase: The radiosensitization effect of x was confirmed in vivo in glioma bearing mice
- Line 192: rephrase: ‘the best’ to highest deltaP50\
- Line 247: rephrase result of Figure 3, add concentration from which it started to be cytotoxic and add statistics (p values). How was the relative cell growth defined? `without selectivity` is not clear?
- 4. needs rephrasing. First results Fig4b and then Fig 4a -> change. Bracket on y-axis of Fig 4a misplaced. Results of Fig 4b are not clearly explained.
- 6 needs rephrasing. Line 315: living animal imaging system? Specify (BLI?). Dynamic changes of ? in the brain tumor? Cobalt 60 source for divided dose? Does this mean a fractionated dose delivery? What was the total dose and dose of the fractions?
- Line 317. Radiation therapy is abbreviated but not used later (e.g. line 318)
- Fig 6: treatment every four days for four consecutive times ? This means 4 times drug with or without irradiation every fourth day? So in total 16 days of therapy?
- Fig 6b = define what is on the graph in the legend (photon flux in function of…)
- Rephrase results of Fig 6 in the text. First mention significant decrease in bioluminescent signal between vehicle versus Efa+IR, IR only and 19c+IR. Then compare IR only versus combined treatments.
- Add absolute p-values throughout the results sections (not only < or > 0.05)
- Figure 7a. X-axis -> tumor inoculation (not plantation). Y-axis: fraction survival?
- Figure 7. Adapt title: survival time extension should not be the start of the title
- ‘Data are’ -> adapt to data represents
- Table 5: abbreviations need to be explained under the table
- Line 350: which suggests that oral administration may be preferred?
CONCLUSION: rephrasing needed
- Line 355-356: `it is particularly important to develop a radiosensitizing agent to avoid the obstacle of the blood-brain barrier` This is not true -> to overcome radioresistance influenced by the presence of hypoxia. BBB passage is necessary to enable the compound to reach the drug and exert its anti-hypoxic effects.
- Line 365: `whose in vitro activities` -> this needs to be specified?
- Line 371: rephrase
Figure S1. Explain abbreviations in legend (IP, PO)
Chemistry:
- Coupling constant errors (e.g. line 613 and 614), check NMR data
- Abbreviations of solvents not added in full (e.g. line 128,129)
- SARs -> SAR
- Line 388-391?
- Line 378, Line 397 (filter cake?)
Reviewer 2 Report
- I recommand to check all the chemical formula, for example LiAlH4, not LiAlH4, DMSO-d6, not DMSO-D6 etc;
- Material and methods 4.2 it is not clear;
- I recommand to add 13C NMR spectra for the final compounds in the Supporting Informations
- Sentence 19 is not clear
- Better justify HEK293 and U81 cell lines for cytotoxic analysis
Reviewer 3 Report
The authors synthesized a series of phenoxyacetic acid derivative based on the efaproxiral structure. Testing these molecules in vitro yielded a handful of compounds had comparable or slightly better allosteric effect on hemoglobin compared to the parent molecule efaproxiral. While the lead molecules showed good cytotoxicity in human cells, the synergic effect with Co-60 irradiation treatment was apparent. The combination 19C + IR provided the best anti-tumor results. Overall, this manuscript provided a full story from chemical analoging to in vitro assays to in vivo study, which consistently suggested that phenoxyacetic acid was a privileged core as hemoglobin allosteric modulator and the new modality that using phenoxyacetic acid derivative to sensitize irradiation therapy was viable. Therefore, I endorse the publication of this manuscript in Molecules after a couple of minor edits.
- Figure 2a, reproduction of a known crystal structure requires noting the PDB entry in the figure caption.
- Scheme 2, substituents on compounds 14a-d are confusing. Group R3 is unknown.
- Figure 3, the cytotoxicity curve suggested that Compounds 19c and 19t were quite soluble in water to nearly 100 mM. This is not common with such organic moieties. I suggested the authors to check the solubilities of these two compounds to make sure that these curves make sense.
- Page 15, Section 4.2, is this section messed up?
- Compound 15a contained quite a couple of impurities according to the NMR. Need to clean up a bit. This purity issue should also have influenced the precision of the data related to 15a. Those data need a re-acquisition.
Round 2
Reviewer 1 Report
The authors addressed most of the comments and this improved the manuscript. However, some comments are outstanding:
- Radiosensitization effect in Figure 6c is not clear
- limitation of not studying the hypoxic effect should be added to the discussion or future perspective
- Figure 4b and explanation is still not clear
- Comment 21:
The radiotherapy was divided into four doses with a total dose of 12Gy. -> rephrase: The tumor was irradiated with a total dose of 12 Gy in four fractions of 3 Gy. Add details of field size of the irradiation and dose rate, keV, filter,…? How did you target the tumor? Image guided irradiation or just total brain irradiation?
- Comment 23:
rephrase: Efaproxiral or 19c was administrated at a dosage of 150mg/kg by intraperitoneal injection (ip) every four days over a period of sixteen days.
- Comment 25:
As shown in the schematic diagram, 19c administration would increase tumor oxygenation in GB. Reoxygenation of hypoxic area sensitized IR treatment
This new sentence is a hypothesis, not a fact since the authors did not confirm changes in hypoxia, only survival and tumor volume so should be deleted in the results section but can be added to the discussion as a hypothesis, not a fact. -
Comment 33: Line 365: `whose in vitro activities` -> this needs to be specified?
Response: We have specified the it in the revised version.
This sentence is still not specific enough: n. In the in vitro activity evaluation stage, we obtained two compounds in vitro we confirmed that both 19c and 19t, whose in vitro activities were had significantly higher activity than that of the positive control -> higher activity, which activity? This should be specified
